# Information-constrained optimization: can adaptive processing of gradients help?

**Jayadev Acharya**[*]
Cornell University
acharya@cornell.edu

**Clément L. Canonne**[†]
University of Sydney
clement.canonne@sydney.edu.au

**Prathamesh Mayekar**[‡]
Indian Institute of Science
prathamesh@iisc.ac.in

**Himanshu Tyagi**[§]
Indian Institute of Science
htyagi@iisc.ac.in

## Abstract

We revisit first-order optimization under local information constraints such as local privacy, gradient quantization, and computational constraints limiting access to a few coordinates of the gradient. In this setting, the optimization algorithm is not allowed to directly access the complete output of the gradient oracle, but only gets limited information about it subject to the local information constraints. We study the role of *adaptivity* in processing the gradient output to obtain this limited information from it, and obtain tight or nearly tight bounds for both convex and strongly convex optimization when adaptive gradient processing is allowed.

## 1  Introduction

Distributed optimization has emerged as a central tool in federated learning for building statistical and machine learning models for data distributed across users. In addition, large scale optimization is typically implemented in a distributed fashion over multiple machines or multiple cores within the same machine. These distributed implementations fit naturally in the oracle framework of first-order optimization (see [22]) where in each iteration a user or machine computes the gradient oracle output. Due to practical local constraints such as communication bandwidth, privacy concerns, or computational issues, the entire gradient cannot be directly made available to the optimization algorithm. Instead, the gradients must be passed through a randomized mechanism which, respectively, ensures privacy of user data (local privacy constraints); or compresses them to a small number of bits (communication constraints); or only computes a few coordinates of the gradient (computational constraints). Motivated by these applications, we consider first-order optimization under such local information constraints placed on the gradient oracle.

When designing a first-order optimization algorithm under local information constraints, one not only needs to design the optimization algorithm itself, but also the algorithm for local processing of the gradient estimates. Many such algorithms have been proposed in recent years; see, for instance, [12], [1], [6], [15], [29], [16], and the references therein for privacy constraints ; [28], [7], [30], [18],

---

[*]Supported by NSF-CCF-1846300 (CAREER), NSF-CCF-1815893.

[†]Part of this work was done while at IBM Research, supported by a Goldstine Postdoctoral Fellowship.

[‡]Supported by a Ph.D. fellowship from Wipro Ltd.

[§]Supported by a grant from Robert Bosch Center for Cyber Physical Systems (RBCCPS), Indian Institute of Science.

The full version of this paper [2] contains the proofs of all results discussed here, as well as other additional details.

35th Conference on Neural Information Processing Systems (NeurIPS 2021).

[14], [25], [4], [11], [17], [20], [19], [27], and the references therein for communication constraints; [24, 26] for computational constraints. However, these algorithms primarily consider *nonadaptive* procedures for gradient processing (with the exception of [14]): that is, where the scheme used to process the gradients at any iteration cannot depend on information gleaned from previous iterations. As a result, the following question remains largely open:

*Can adaptively processing gradients improve convergence in information-constrained optimization?*

In this paper, we study this question for optimization over both convex and strongly convex function families and under the three different local constraints mentioned above: local privacy, communication, and computational. For each of these constraints, we establish lower bounds on convergence rates which hold even when the gradients are adaptively processed. In the next sections, we cover prior related work, before elaborating on our results and techniques.

## 1.1 Prior work

The model studied here can be viewed as extension of the classical query complexity model in [22]. Without information constraints, [5] study this setting and provide a general recipe for proving convex optimization lower bounds for different function families. Specifically, they reduce the problem of optimization with a first-order oracle to a mean estimation problem whose probability of error is lower bounded using Fano's method ($cf.$ [31]). While our work also relies on a reduction to mean estimation, we deviate from their approach, using instead Assouad's method to prove lower bounds for various function families. This different approach in turn enables us to derive lower bounds for adaptive processing of gradients.

In the information-constrained setting, motivated by privacy concerns, [12] consider the problem where the gradient estimates must pass through a locally differentially private (LDP) channel. However, in their setting the LDP channels for all time steps are selected at the start of optimization algorithm – in other words, the channel selection strategy is nonadaptive. Similarly, [20] and [19] consider a similar problem and impose the constraint that the gradient estimates be quantized to a fixed number of bits. They, too, fix the quantization channels used at each time step at the start of optimization algorithm. In contrast, in this paper, we allow for *adaptive* channel selection strategies; as a result, the lower bounds established in these papers do not apply to our setting, and are more restrictive than our bounds.

The results of Duchi and Rogers [13] for Bernoulli product distributions could be combined with our construction to obtain tight lower bounds for optimization in $p \in [1, 2]$ under LDP constraints, but would not extend to the entire range of $p$. The work of Braverman, Garg, Ma, Nguyen, and Woodruff [9] on communication constraints, also for $p \in [1, 2]$, is relevant as well; however, their bounds on mutual information cannot be applied directly, as their setting (Gaussian distributions) would not satisfy our almost sure gradient oracle assumption.

[14] provide adaptive quantization schemes for convex and $\ell_2$ Lipschitz function family. While the worst-case convergence guarantees for the quantizers in [14] are similar to those in [7] and [20], it shows some practical improvements over the state-of-the-art for some specific problem instances. This suggests that while adaptive quantization may not help in the worst case for non-smooth convex optimization, it may be useful for a smaller subclass of convex optimization problems.

## 1.2 Our contributions

We consider a general formulation for first-order optimization under local information constraints where the output of the gradient oracle must be passed through a channel $W$ selected from a fixed channel family $\mathcal{W}$. This family $\mathcal{W}$ captures the information constraints; see Section 2.3 for a description of the channel families corresponding to our constraints of interest. Specifically, the gradient is sent as input to this channel $W$, and the algorithm receives the output. In each iteration of the algorithm, the channel to be used in that iteration can be selected adaptively based on previously received channel outputs by the algorithm or channels to be used throughout can be fixed upfront, nonadaptively. The detailed problem setup is given in Section 2.1. We obtain general lower bounds for optimization of convex and strongly convex functions using $\mathcal{W}$, when adaptivity is allowed. These bounds are then applied to the specific constraints of interest to obtain our main results.

Our first contribution is in showing that adaptive gradient processing does *not* help for some of the most typical optimization problems. Namely, we prove that for most regimes of local privacy, communication, or computational constraints, adaptive gradient processing has nearly the same convergence rate as nonadaptive gradient processing for both convex and strongly convex function families. As a consequence, this shows that the nondaptive LDP algorithms from [12] and nonadaptive compression protocols from [20], [19] are optimal for private and communication-constrained optimization, respectively. In another direction, under computational constraints, where we are allowed to compute only one gradient coordinate, we show that standard RCD ($cf.$ [10, Section 6.4]), which employs uniform (nonadaptive) sampling of gradient coordinates, is optimal for convex and strongly convex function families. This proves that adaptive sampling of gradient coordinates does not improve over nonadaptive sampling strategies.

As previously discussed, prior work in both the local privacy and communication-constrained settings concerned itself with the family of convex functions, with no lower bounds known for the more restricted family of *strongly convex* functions, even for nonadaptive gradient processing protocols. The key obstacle is the fact that during the reduction from optimization to mean estimation, the known difficult case for the strongly convex family, even when analyzed for nonadaptive protocols, leads to an estimation problem using adaptive protocols, and the lack of known lower bounds for adaptive information-constrained estimation prevented this approach from succeeding. Specifically, this difficult case has gradients that can depend on the query point which in turn can be chosen based on previously observed channel outputs, an issue which does not arise in the case of the convex family, where the lower bounds are derived using affine functions for which the gradients do not depend on query point. We manage to circumvent this issue, by relying on a different reduction which lets us capitalize on a recent lower bound for adaptive mean estimation. Crucially, this recent lower bound does apply to adaptive estimation algorithms as well, bringing the last missing piece to the puzzle. This lets us derive lower bounds for both convex Lipschitz *and* strongly convex functions under adaptive gradient processing.

For general function classes, the results discussed above show that adaptive processing of gradients does not help. This begs the question of whether there are natural function families where adaptive gradient processing can lead to significant savings. Our third contribution is to provide an example of such a family: specifically, we exhibit a natural optimization problem (entailing $\ell_2$ minimization) under computational constraints for which adaptive gradient processing provides a polynomial factor improvement in convergence rates compared to nonadaptive processing. The key feature of this optimization problem is that the resulting gradients have structured sparsity; adaptivity then allows for a two-phase optimization procedure, where the algorithm first "explores" to find the structure before, in a second phase, "exploiting" it to obtain more focused information about the function to minimize. However, nonadaptive gradient processing protocols cannot easily exploit this hidden structure, as finding it is now akin to locating a needle in a haystack; and thus exhibit much slower convergence rates.

**Notation**: Throughout the paper $q$ denotes the Hölder conjugate of $p$ (that is, $q = p/(p-1)$). $a \vee b$ and $a \wedge b$ denote... $\max\{a, b\}$ and $\min\{a, b\}$, respectively. We denote by $\log$ the logarithm to the base 2 and by $\ln$ the logarithm to the base $e$. The information-theoretic quantities, such as mutual information and KL divergence, are defined using $\ln$. $\ln^*(a)$ denotes the number of times $\ln$ must be iteratively applied to $a$ before the result is at most 1. $\{e_1, ..., e_d\}$ denotes the standard basis of $\mathbb{R}^d$.

## 2 Setup and preliminaries

### 2.1 Optimization under information constraints

We consider the problem of minimizing an unknown convex function $f \colon \mathcal{X} \to \mathbb{R}$ over its domain $\mathcal{X}$ using *oracle access* to noisy subgradients of the function ($cf.$ [22]). In our setup, the gradient estimates supplied by the oracle must pass through a channel $W$,[5] chosen from a fixed set of channels $\mathcal{W}$, and the optimization algorithm $\pi$ only has access to the output of this channel. The *channel family* $\mathcal{W}$ represents our information constraints.

---

[5] A channel $W$ with input alphabet $\mathcal{X}$ and output alphabet $\mathcal{Y}$, denoted $W \colon \mathcal{X} \to \mathcal{Y}$, represents the conditional distribution of the output of a randomized function given its input. In particular, $W(\cdot|x)$ is the conditional distribution of the channel given that the input is $x \in \mathcal{X}$.

In detail, the framework is as follows:

1. At iteration $t$, the optimization algorithm $\pi$ makes a query for point $x_t$ to the oracle $O$.
2. Upon receiving the point $x_t$, the oracle outputs $\hat{g}(x_t)$, where $\mathbb{E}\left[\hat{g}(x_t) \mid x_t\right] \in \partial f(x_t)$ and $\partial f(x_t)$ is the subgradient set of function $f$ at $x_t$.
3. The subgradient estimate $\hat{g}(x_t)$ is passed through a channel $W_t \in \mathcal{W}$, and the output of the channel $W_t$ along with outputs of channels $\{W_i\}_{i \in [t-1]}$, where $[n] = \{1, \ldots, n\}$, can be used by the first order optimization algorithm to further update $x_t$ to $x_{t+1}$.

Denote by $\pi$ the first-order optimization algorithm which is allowed $T$ queries to the oracle $O$ and, after the $t$th query, gets back the output $Y_t$ with distribution $W_t(\cdot \mid \hat{g}(x_t))$. Denote the set of all such optimization algorithms by $\Pi_t$.

Our goal is to select channels $W_t$s and an optimization algorithm $\pi$ to get a small worst-case optimization error. Two classes of *channel selection strategies* are of interest: *adaptive* and *nonadaptive*.

Adaptive channel selection strategies can be described as follows. The channel $W_t$ selected at time $t$ may depend on the previous outputs of channels $\{W_i\}_{i \in [t-1]}$. Specifically, denoting by $\mathcal{Y}_t$ and $Y_t$, respectively, the output alphabet and the output for the channel used at time $t$, the *adaptive channel selection strategy* $S := (S_1, \ldots, S_T)$ over $T$ iterations comprises mappings $S_t \colon \mathcal{Y}^{t-1} \to \mathcal{W}$ taking $Y_1, \ldots Y_{t-1}$ as input and yielding a channel in $\mathcal{W}$ as output. We write $\mathcal{S}_{\mathcal{W},T}$ for the collection of all such channel selection strategies.

For nonadaptive channel selection strategies, we fix the channels $\{W_t\}_{t \in [T]} \in \mathcal{W}_{\text{obl}}$ through which all the gradient estimates must pass at the start of the optimization algorithm.[6] Denote the class of all nonadaptive strategies by $\mathcal{S}_{\mathcal{W},T}^{\text{NA}}$.

We measure the performance of an optimization protocol $\pi$ and a channel selection strategy $S$ for a given function $f$ and oracle $O$ using the metric $\mathcal{E}(f, O, \pi, S)$ defined as

$$\mathcal{E}(f, O, \pi, S) := \mathbb{E}\left[f(x_T) - \min_{x \in \mathcal{X}} f(x)\right], \tag{1}$$

where the expectation is over the randomness in $x_T$.

For various function and oracle classes, denoted by $\mathcal{O}$, the channel constraint family $\mathcal{W}$, and the number of iteration $T$, we will characterize the *adaptive minmax optimization error*

$$\mathcal{E}^*(\mathcal{X}, \mathcal{O}, T, \mathcal{W}) = \inf_{\pi \in \Pi_T} \inf_{S \in \mathcal{S}_{\mathcal{W},T}} \sup_{(f,O) \in \mathcal{O}} \mathcal{E}(f, O, \pi, S) \tag{2}$$

and the corresponding *nonadaptive minmax optimization error*

$$\mathcal{E}^{\text{NA}*}(\mathcal{X}, \mathcal{O}, T, \mathcal{W}) = \inf_{\pi \in \Pi_T} \inf_{S \in \mathcal{S}_{\mathcal{W},T}^{\text{NA}}} \sup_{(f,O) \in \mathcal{O}} \mathcal{E}(f, O, \pi, S). \tag{3}$$

Since nonadaptive channel selection strategies are a subset of adaptive channel selection strategies, we have $\mathcal{E}^{\text{NA}*}(\mathcal{X}, \mathcal{O}, T, \mathcal{W}) \geq \mathcal{E}^*(\mathcal{X}, \mathcal{O}, T, \mathcal{W})$.

## 2.2 Function classes

We define the function classes we consider, and a structured optimization problem for which we show that adaptivity in the choice of channels helps.

**Convex and $\ell_p$ Lipschitz function family.** Our first set of function families are parameterized by a number $p \in [1, \infty]$. Throughout, we restrict ourselves to convex functions over a domain $\mathcal{X}$, i.e.,

$$f(\lambda x + (1 - \lambda)y) \leq \lambda f(x) + (1 - \lambda)f(y), \quad \forall x, y \in \mathcal{X}, \quad \forall \lambda \in [0, 1]. \tag{4}$$

Further, for a family parameterized by $p$, we assume that the subgradient estimates returned by the first-order oracle for a function $f$ satisfy the following two assumptions:

$$\mathbb{E}\left[\hat{g}(x) \mid x\right] \in \partial f(x), \quad \text{(Unbiased estimates)} \tag{5}$$

$$\Pr\left(\|\hat{g}(x)\|_q^2 \leq B^2 \mid x\right) = 1, \quad \text{(Bounded estimates)} \tag{6}$$

---

[6]That is, $W_t$ is selected independently of the $t - 1$ gradient observations received by the optimization algorithm at step $t$.

where $\partial f(x)$ is the set of subgradient for $f$ at $x$ and $q := p/(p-1)$ is the Hölder conjugate of $p$. We denote by $\mathcal{O}_{\mathtt{c},p}$ the set of all pairs of functions and oracles satisfying assumptions (4), (5), and (6).

We note that (5) is standard in stochastic optimization literature (*cf.* [22], [21], [10], [5]). To prove convergence guarantees on first-order optimization in the classic setup (without any information constraints on the oracle), it is enough to assume $\mathbb{E}\left[\|\hat{g}(x)\|_q^2\right] \leq B^2$. We make a slightly stronger assumption in this case since the more relaxed assumption leads to technical difficulties in finding unbiased quantizers for gradients; see [20, 19]. Also, observe that assumption (6) imposes a restriction on the functions allowed in this class. Note that by (5) and (6) for every $x \in \mathcal{X}$ there exists a vector $g \in \partial f(x)$ such that $\|g\|_q \leq B$. Further, since $f$ is convex, $f(x) - f(y) \leq g^T(x-y)$ for every $g \in \partial f(x)$, whereby $|f(x) - f(y)| \leq B\|x-y\|_p$. Namely, $f$ is $B$-Lipschitz continuous in the $\ell_p$ norm.[7]

**Remark 1** (Convergence rate for convex functions). *Without any information constraints (when gradient estimates are directly observed), upper bounds of $\frac{c_1 DB\sqrt{\log d}}{\sqrt{T}}$ and $\frac{c_1 DBd^{1/2-1/p}}{\sqrt{T}}$ on the error are achievable for $\ell_1$ and $\ell_p$, $p \in [2,\infty]$, convex family, respectively. Moreover, these rates are orderwise optimal. In particular, from [5, Appendix C] we have the following result: For $p = 1$, stochastic mirror descent algorithm with mirror map $\Phi_a(x) = \frac{1}{a-1}\|x\|_a^2$, where $a = \frac{2\log d}{2\log d - 1}$, achieves the orderwise convergence rate; for $p \in [2,\infty]$, stochastic gradient descent achieves the orderwise optimal convergence rate.*

**Strongly convex and $\ell_2$ Lipschitz function family.** We now consider a special subset of the convex and $\ell_2$ Lipschitz family described above, where the functions are strongly convex. For $\alpha > 0$, a function $f$ is $\alpha$-*strongly convex on* $\mathcal{X}$ if the function $h$ defined below is convex:

$$h(x) = f(x) - \frac{\alpha}{2}\|x\|_2^2, \quad \forall x \in \mathcal{X}. \tag{7}$$

We denote by $\mathcal{O}_{\mathtt{sc}}$ the set of all pairs of functions and oracles satisfying (4), (5), (6), and (7), where (6) is satisfied for $q = 2$.

The strong convexity parameter $\alpha$ is related to the parameter $B$, the upper bound on the $\ell_2$ norm of the gradient estimate. We state a relation between them when the domain $\mathcal{X}$ contains an $\ell_\infty$ ball of radius $D$ centered at the origin, as this will be of interest while proving lower bounds.

**Lemma 1.** *For any $\mathcal{X} \supseteq \{x : \|x\|_\infty \leq D\}$, we have $\frac{B}{\alpha} \geq \frac{Dd^{1/2}}{4}$.*

**Remark 2** (Convergence rate for strongly convex functions). *Without information constraints, stochastic gradient descent achieves an upper bound of $\frac{2B^2}{T+1}$ ([21]) for the strongly convex family, and this rate is optimal; see [5].*

**Mean estimation as an optimization problem.** Our final optimization problem entails a structured $\ell_2$ minimization problem. We first define $s$-block sparsity which is needed for our function classes.

**Definition 1.** *A vector $v \in \mathbb{R}^d$ is $s$-block sparse if (i) there exists an $i$ such that $v_j = 0$ for all $j \notin \{is+1, \ldots, \min\{i(s+1), d\}\}$ and (ii) the non-zero coordinates have the same absolute value in $[0,1]$. Let $\mathcal{B}_s$ be the set of all $s$-block sparse vectors in $d$ dimensions.*[8]

For $v \in \mathcal{B}_s$, we define a function $f_v$ over $\mathcal{X} = [-1,1]^d$ as $f_v(x) = \|x - v\|_2^2, x \in \mathcal{X}$. Further, we associate with each function $f_v$ an oracle $O_v$ as follows. Let $X$ be a random variable over $\{-1,1\}^d$ with $\mathbb{E}[X] = v$ (i.e., its mean is the $s$-block sparse vector $v$ parameterizing $f_v$). The gradient estimate output of the oracle $O_v$ at $x$ and at time $t$ is $2(x - X_t)$, where $\{X_t\}_{t=1}^\infty$ are i.i.d. random variables with the same distribution as $X$. Note that the expected value of this gradient estimate is $\nabla f(x)$.

**Definition 2.** *Let $\mathcal{O}_{\mathtt{blsp},s}$ denote the collection of pairs of functions and oracles described above.*

Observe that the first-order optimization described above is the standard $\ell_2$ mean estimation problem cast as an optimization problem, since the function $f_v$ is minimized at $x^* := \mathbb{E}[X]$. Moreover, the essential information supplied by the oracle are the i.i.d. samples $X_t$ (since the optimization algorithm already knows the queries $x$).

---

[7]The same could be said under the weaker assumption $\mathbb{E}\left[\|\hat{g}(x)\|_q^2\right] \leq B^2$.

[8]We assume throughout, for simplicity, that $d/s$ is an integer.

## 2.3  Information constraints

We describe three specific constraints of interest to us: local privacy, communication, and computation. The first two are well-studied; the third arises in procedures such as random coordinate descent.

**Local differential privacy.** To model local privacy, we define the $\varepsilon$-locally differentially private (LDP) channel family $\mathcal{W}_{\mathtt{priv},\varepsilon}$.

**Definition 3.** *A channel* $W : \mathbb{R}^d \to \mathbb{R}^d$ *is $\varepsilon$-locally differentially private ($\varepsilon$-LDP) if for all* $x, x' \in \mathbb{R}^d$,

$$\frac{W(Y \in S \mid X = x)}{W(Y \in S \mid X = x')} \le e^{\varepsilon}$$

*for all Borel measurable subsets $S$ of $\mathbb{R}^d$. We denote by $\mathcal{W}_{\mathtt{priv},\varepsilon}$ the set of all $\varepsilon$-LDP channels.*

When operating under local privacy constraints, oracle's subgradient estimates are passed through a $\varepsilon$-LDP channel, and only the output is available to the optimization algorithm. Thus, the data of individual users, accessed in each oracle query, remains differentially private, a notion of privacy that is now widely agreed upon.

**Communication constraints.** To model communication constraints, we define the $\mathcal{W}_{\mathtt{com},r}$ the $r$-bit communication-constrained channel family, as follows.

**Definition 4.** *A channel* $W : \mathbb{R}^d \to \{0,1\}^r$ *constitutes an $r$-bit communication-constrained* channel. *We denote by $\mathcal{W}_{\mathtt{com},r}$ the set of all $r$-bit communication-constrained channels.*

**Computational constraints.** For high-dimensional optimization, altogether computing the subgradient estimates can be computationally expensive. Often in such cases, one resorts to computing only a few coordinates of the gradient estimates and using only them for optimization ([24, 26]). This motivates the oblivious sampling channel family $\mathcal{W}_{\mathtt{obl}}$, where the optimization algorithm gets to see only one randomly chosen coordinate of the gradient estimate.

**Definition 5.** *An* oblivious sampling *channel $W$ is a channel* $W : \mathbb{R}^d \to \mathbb{R}^d$ *specified by a probability vector $(p_i)_{i \in [d]}$, i.e., a vector $p$ such that $p_i \ge 0$ for all $i$ and $\sum_{i \in [d]} p_i = 1$. For an input $g \in \mathbb{R}^d$, the output distribution of $W$ is given by $W(g(i)e_i \mid g) = p_i, \forall i \in [d]$. We denote by $\mathcal{W}_{\mathtt{obl}}$ the set of all oblivious sampling channels.*

Therefore, at most one coordinate of the oracle's the gradient estimate can be used by the optimization algorithm. Further, this coordinate is sampled obliviously to the input gradient estimate itself. We note that the special case of $p = \frac{1}{d}\mathbf{1}_d$ corresponds to sampling employed by standard *Random Coordinate Descent* (RCD) (*cf.* [10, Section 6.4]), where at each time step only one uniformly random coordinate of the gradient is used by the gradient descent algorithm.

## 3  Main results

For $p \in [1, \infty]$ and $D > 0$, let $\mathbb{X}_p(D) := \{\mathcal{X} \subseteq \mathbb{R}^d : \max_{x,y \in \mathcal{X}} \|x - y\|_p \le D\}$ be the collection of subsets of $\mathbb{R}^d$ whose $\ell_p$ diameter is at most $D$. In stating our results, we will fix throughout the parameter $B > 0$, the almost sure bound on the gradient magnitude defined in (6), as well as the strong convexity parameter $\alpha > 0$ defined in (7) (which, implicitly, is required to satisfy Lemma 1). Throughout this section, our lower bounds on minmax optimization error focus on tracking the convergence rate for large $T$, a standard regime of interest for the stochastic optimization setting.

### 3.1  Lower bounds on locally private optimization under adaptive gradient processing

Throughout, we consider $\varepsilon \in [0, 1]$, namely the high-privacy regime.

**Convex function family.** For the convex function family, we prove the following lower bounds.

**Theorem 1.** *Let $p = 1, \varepsilon \in [0, 1]$, and $D > 0$. There exists a constant $c_0$ such that for $T = \Omega(d/\varepsilon^2)$,*

$$\sup_{\mathcal{X} \in \mathbb{X}_1(D)} \mathcal{E}^*(\mathcal{X}, \mathcal{O}_{\mathtt{c},1}, T, \mathcal{W}_{\mathtt{priv},\varepsilon}) \ge \frac{c_0 DB}{\sqrt{T}} \cdot \sqrt{\frac{d}{\varepsilon^2}}.$$

**Theorem 2.** *Let $p \in [2, \infty], \varepsilon \in [0, 1]$, and $D > 0$. There exists a constant $c_0$ such that for $T = \Omega(d^2/\varepsilon^2)$,*

$$\sup_{\mathcal{X} \in \mathbb{X}_p(D)} \mathcal{E}^*(\mathcal{X}, \mathcal{O}_{\mathtt{c},p}, T, \mathcal{W}_{\mathtt{priv},\varepsilon}) \geq \frac{c_0 DB d^{1/2-1/p}}{\sqrt{T}} \cdot \sqrt{\frac{d}{\varepsilon^2}}.$$

**Remark 3** (Tightness of bounds for convex functions and LDP constraints). *[12, Theorem 4 and 5] provide nonadaptive LDP algorithms which show that Theorem 1 is tight up to logarithmic factors for $p = 1$ and Theorem 2 is tight up to constant factors for all $p \in [2, \infty]$. Therefore, adaptive processing of gradients under LDP cannot significantly improve the convergence rate for convex function families.*

*Interestingly, for $p = 1$, [12] also provide a slightly stronger lower bound of $\frac{c_0 DB}{\sqrt{T}} \cdot \sqrt{\frac{d \log d}{\varepsilon^2}}$ for nonadaptive protocols, which matches the performance of their nonadaptive protocols up to constant factors. This points to a minor gap in our understanding of adaptive protocols: Can we establish a stronger lower bound for adaptive protocols to match the performance of the nonadaptive algorithm of [12], or does there exist a better adaptive protocol? We believe that the latter option is correct, and conjecture that the $\sqrt{d \log d}$ dependence is tight even for adaptive protocols.*

From Remark 1, the standard optimization error for $\ell_1$ and $\ell_p$, $p \in [2, \infty]$, convex family blows up by a factor of $\sqrt{\frac{d}{\varepsilon^2}}$ when the gradient estimates are passed through an $\varepsilon$-LDP channel.

**Remark 4.** *We note that our techniques also yield lower bounds for $p \in (1, 2)$, a range that has not been considered in prior works on information-constrained gradient processing to the best of our knowledge. These bounds are given in the full version [2]. Deriving tight upper bounds for this range is an interesting open question, which we leave for future exploration.*

**Strongly convex family.** We prove the following result for strongly convex functions.

**Theorem 3.** *Let $p = 2$, $\varepsilon \in [0, 1]$, and $D > 0$. There exists a constant $c_0 > 0$ such that for $T \geq \Omega\left(\frac{B^2}{\alpha^2 D^2} \cdot \frac{d}{\varepsilon^2}\right)$,*

$$\sup_{\mathcal{X} \in \mathbb{X}_2(D)} \mathcal{E}^*(\mathcal{X}, \mathcal{O}_{\mathtt{sc}}, T, \mathcal{W}_{\mathtt{priv},\varepsilon}) \geq \frac{c_0 B^2}{\alpha T} \cdot \frac{d}{\varepsilon^2}.$$

**Remark 5** (Tightness of bounds for strongly convex functions and LDP constraints). *One can use stochastic gradient descent with the nonadaptive protocol from [12, Appendix C.2] to obtain a nonadaptive protocol with convergence rate matching the lower bound in Theorem 3 up to constant factors, establishing that adaptivity does not help for strongly convex functions.*

From Remark 2, the standard optimization error for strongly convex functions blows up by a factor of $\frac{d}{\varepsilon^2}$ when the gradient estimates are passed through an $\varepsilon$-LDP channel.

### 3.2 Lower bounds on communication-constrained optimization

**Convex function family.** For convex functions we prove the following lower bounds.

**Theorem 4.** *Let $p = 1$. There exists a constant $c_0 > 0$ such that for $r \in \mathbb{N}$, $T = \Omega(d/r)$, and $D > 0$*

$$\sup_{\mathcal{X} \in \mathbb{X}_1(D)} \mathcal{E}^*(\mathcal{X}, \mathcal{O}_{\mathtt{c},1}, T, \mathcal{W}_{\mathtt{com},r}) \geq \frac{c_0 DB}{\sqrt{T}} \cdot \sqrt{\frac{d}{d \wedge r}}.$$

**Theorem 5.** *Let $p \in [2, \infty]$. There exists a constant $c_0 > 0$ such that for $r \in \mathbb{N}$, and $T = \Omega\left(\frac{d^2}{2^r \wedge d}\right)$, and $D > 0$*

$$\sup_{\mathcal{X} \in \mathbb{X}_p(D)} \mathcal{E}^*(\mathcal{X}, \mathcal{O}_{\mathtt{c},p}, T, \mathcal{W}_{\mathtt{com},r}) \geq \left(\frac{c_0 DB d^{1/2-1/p}}{\sqrt{T}} \cdot \sqrt{\frac{d}{d \wedge 2^r}}\right) \vee \left(\frac{c_0 DB}{\sqrt{T}} \cdot \sqrt{\frac{d}{d \wedge r}}\right).$$

**Remark 6** (Tightness of bounds for convex functions and communication constraints). *In the full version of the paper [2], we provide a scheme which matches the lower bound in Theorem 4 up to constant factors for any $r$. Since each coordinate of oracle output is bounded by $B$ for $p = 1$, we*

*simply can use an unbiased 1-bit quantizer for each coordinate. The proposed scheme uses such a quantizer for each coordinate and makes $d/r$ repeated queries to the oracle for the same point, but gets 1-bit information about $r$ different coordinates in each query.*

*For $p \in [2, \infty]$, we can use the quantizer $\mathsf{SimQ}^+$ from [19] with $k = r$ and appropriate mirror descent algorithms to get upper bounds that match the lower bounds in Theorem 5, up to an additional $O(\log d)$ factor. For $p = 2$, we can use the quantizer $\mathsf{RATQ}$ from [20] to improve this match to an $O(\ln \ln^* d)$ factor.*

From Remark 1, the standard optimization errors for $\ell_1$ and $\ell_p$, $p \in [2, \infty]$, convex family blow up by a factor of $\sqrt{\frac{d}{d \wedge r}}$ and $\sqrt{\frac{d}{d \wedge 2^r}} \vee \sqrt{\frac{d^{2/p}}{d \wedge r}}$, respectively, when the gradient estimates are compressed to $r$ bits.

**Remark 7.** *Finally, we remark that our techniques also extend to lower bounds for communication-constrained optimization of $\ell_p$, $p \in (1, 2)$, convex family, which we prove in the proofs section, but the known upper bounds in the previous works are not tight for this family for $r \leq d$.*

**Strongly convex family.** We prove the following result for strongly convex functions.

**Theorem 6.** *Let $p = 2$ and $D > 0$. There exists a constant $c_0 > 0$ such that, for $r \in \mathbb{N}$ and $T = \Omega\left(\frac{B^2}{\alpha^2 D^2} \cdot \frac{d}{r}\right)$,*

$$\sup_{\mathcal{X} \in \mathbb{X}_2(D)} \mathcal{E}^*(\mathcal{X}, \mathcal{O}_{\mathsf{sc}}, T, \mathcal{W}_{\mathsf{com}, r}) \geq \frac{c_0 B^2}{\alpha T} \cdot \frac{d}{d \wedge r}.$$

**Remark 8** (Tightness of bounds for strongly convex functions and communication constraints)**.** *We note that the nonadaptive scheme $\mathsf{RATQ}$ in [20] along with stochastic gradient descent matches the lower bound in Theorem 6 up to a $\ln \ln^* d$ factor for $r = \Omega(\ln \ln^* d)$.*

From Remark 2, the standard optimization error for strongly convex functions blows up by a factor of $\frac{d}{r}$ when the gradient estimates are compressed to $r$ bits.

### 3.3 Lower bounds on computationally-constrained optimization

Our motivation for studying the oblivious sampling channel is to derive lower bounds for random coordinate descent methods. Since such optimization methods are natural to the $\ell_2$ space, we restrict our attention to the convex family and $\ell_2$ Lipschitz family in this section.

**Theorem 7** (Convex family)**.** *Let $p = 2$. There exists a constant $c_0 > 0$ such that, for $T = \Omega(d)$ and $D > 0$,*

$$\sup_{\mathcal{X} \in \mathbb{X}_2(D)} \mathcal{E}^*(\mathcal{X}, \mathcal{O}_{\mathsf{c},2}, T, \mathcal{W}_{\mathsf{obl}}) \geq \frac{c_0 \sqrt{d} D B}{\sqrt{T}}.$$

The standard Random Coordinate Descent (RCD) (see for instance [10, Theorem 6.6]), which employs uniform sampling, matches this lower bound up to constant factors. The optimality of standard RCD reinforces the folklore approach of uniformly sampling coordinates for random coordinate descent unless there is an obvious structure to exploit (as in [23]). This establishes that adaptive sampling strategies do not improve over nonadaptive sampling strategies for the family $\mathcal{W}_{\mathsf{obl}}$. Also from Remark 1, the standard optimization error for $\ell_2$ convex family blows up by a factor of $\sqrt{d}$ when the gradients are sampled obliviously.

**Theorem 8** (Strongly convex family)**.** *Let $p = 2$. There exists a constant $c_0 > 0$ such that for $T = \Omega(d \frac{B^2}{\alpha^2 D^2})$, and $D > 0$*

$$\sup_{\mathcal{X} \in \mathbb{X}_2(D)} \mathcal{E}^*(\mathcal{X}, \mathcal{O}_{\mathsf{sc}}, T, \mathcal{W}_{\mathsf{obl}}) \geq \frac{c_0 d B^2}{\alpha T}.$$

Once again, the standard RCD algorithm matches this lower bound, which shows that adaptive sampling strategies do not improve over nonadaptive sampling strategies for strongly convex optimization. Further, from Remark 2, the standard optimization error for strongly convex family blows up by a factor of $\sqrt{d}$ when the gradients are sampled obliviously.

### 3.4 An example where adaptivity helps

Until now we showed that for information-constrained first-order optimization over the standard function and oracle classes using popular channel families, adaptive channel selection strategies offer no better convergence guarantees than nonadaptive strategies.

We will consider the function and oracle class as $\mathcal{O}_{\mathtt{blsp},s}$ (Definition 2) using information constraint family as $\mathcal{W}_{\mathtt{obl}}$ (Definition 5) and show that adaptive channel selection strategies strictly outperform the nonadaptive ones. Towards that, we first derive a lower bound for nonadaptive strategies, and then we present an adaptive scheme which improves over this bound.

Recall that $\mathcal{E}^{\mathrm{NA}*}(\mathcal{X}, \mathcal{O}_{\mathtt{blsp},s}, T, \mathcal{W}_{\mathtt{obl}}) \geq \mathcal{E}^*(\mathcal{X}, \mathcal{O}_{\mathtt{blsp},s}, T, \mathcal{W}_{\mathtt{obl}})$. We will show a *strict* separation between the two quantities: for $s := \sqrt{d}$, the error incurred by any nonadaptive strategy is at least $\Omega(d^{3/2}/T)$, while there exists an adaptive strategy achieving error $O((d \log d)/T)$.

**Lower bound for nonadaptive channel selection strategies.** We show the following.

**Theorem 9.** *Let $\mathcal{X} = [-1, 1]^d$. Then, there exists absolute constants $c_0, c_1, c_2 > 0$, such that for any $s \geq c_0$ and $T \geq c_1 d$, we have*

$$\mathcal{E}^{\mathrm{NA}*}(\mathcal{X}, \mathcal{O}_{\mathtt{blsp},s}, T, \mathcal{W}_{\mathtt{obl}}) \geq c_2 \cdot \frac{sd}{T}.$$

**An upper bound for adaptive channel selection strategies.** We now prove a $O((d \log(d/s) + s^2)/T)$ upper bound on the error for adaptive strategies, by exhibiting an adaptive channel selection strategy and optimization procedure we term *Adaptive Coordinate Descent* (ACD), denoted $\pi_{\mathtt{ACD}}$.

First, note that the only new information that the oracles present at each iteration is about the random variable $X$ with $\mathbb{E}[X] = v$ underlying the oracle associated with some function $f_v(x) = \|x - v\|_2^2$ in our family $\mathcal{O}_{\mathtt{blsp},s}$. Thus, the problem at hand becomes that of estimating the mean $v$ using independent copies of $X$. See the supplemental for a detailed description.

Keeping this in mind, our adaptive channel selection strategy is divided in two phases, each making $T/2$ queries to the oracle: the *exploration phase* and the *exploitation phase*. In the exploration phase, we select each block's first coordinate as a representative coordinate for that block and query each representative coordinate $Ts/(2d)$ times. At the end of this phase, an estimate of the mean is formed for each representative coordinate. Next, we select the block whose representative coordinate has the sample mean with the highest absolute value. Then, in the exploitation phase each coordinate of the selected block is queried $T/(2s)$ times. Our optimization algorithm estimates the means of coordinates in the selected block using the sample mean of the values received in the exploitation phase. For the rest of the coordinates, the mean estimate is zero. Finally, our algorithm returns the overall estimated mean vector as the estimated minimizer of the function.

Recall that in RCD, the oracle returns the gradient along a randomly chosen coordinate. In contrast, ACD gets gradient for a particular coordinate in each round, and the choice of the coordinates used in the exploitation phase depends on the observations of the exploration phase. Also, we note that it is possible to interpret our procedure as a coordinate descent algorithm. However, for the ease of presentation, we simply retain the form above. The performance of $\pi_{\mathtt{ACD}}$ is characterized below.

**Theorem 10.** *Fix any $1 \leq s \leq d$, and $(f, O) \in \mathcal{O}_{\mathtt{blsp},s}$.[9] Let $\hat{Y} \in \mathbb{R}^d$ be the point returned by $\pi_{\mathtt{ACD}}$ after $T$ oracle queries to $O$. Then,*

$$\mathbb{E}[f(\hat{Y})] \leq \frac{36 d \ln \frac{d}{s} + 2s^2}{T}.$$

## 4 Sketch of proof for our lower bounds

The proofs of all our lower bounds follow the same general template, summarized below.

**Step 1. Relating optimality gap to average information:** We consider a family of functions $\mathcal{G} = \{g_v : v \in \{-1, 1\}^d\}$ satisfying suitable conditions and associate with it a "discrepancy metric"

---

[9] That is, $f(x) = f_v(x) = \|x - v\|^2$ for some $v$ with block sparsity structure and $O$ gives independent copies of random variable $X$ with $\mathbb{E}[X] = v$.

$\psi(\mathcal{G})$ that allows us to relate the optimality gap of any algorithm to an average mutual information quantity. Specifically, for $V$ distributed uniformly over $\{-1,1\}^d$, we show that the output $x_T$ of any optimization algorithm satisfies[10]

$$\mathbb{E}[g_V(x_T) - \min_{x \in \mathcal{X}} g_V(x)] \geq \frac{d\psi(\mathcal{G})}{6}\Big(1 - \Big(\frac{1}{d}\sum_{i=1}^{d} I\big(V(i) \wedge Y^T\big)\Big)^{1/2}\Big),$$

$Y_t$ is the channel output for the gradient in the $t$th iteration and $Y^T := (Y_1, \ldots, Y_T)$. Heuristically, we have related the gap to optimality to the difficulty of inferring $V$ by observing $Y_1, \ldots, Y^T$. We note that the bound above is similar to that of [5], but instead of mutual information $I\big(V \wedge Y^T\big)$ we get average mutual information per coordinate. This latter quantity is amenable to analysis for adaptive protocols.

**Step 2. Average information bounds:** To bound the average mutual information per coordinate, $\frac{1}{d}\sum_{i=1}^{d} I\big(V(i) \wedge Y^T\big)$, we take recourse to recently proposed bounds from [3]. These bounds hold for $Y^T$ which is output of adaptively selected channels from a fixed channel family $\mathcal{W}$, with i.i.d. input $X^T = (X_1, \ldots, X_T)$ generated from a family of distributions $\{\mathbf{p}_v, v \in \{-1,1\}^d\}$. We view the output of oracle as inputs $X^T$ and derive the required bound. While results in [3] provided bounds for $\mathcal{W}_{\mathtt{priv},\varepsilon}$ and $\mathcal{W}_{\mathtt{comm},r}$, we extend the approach to handle $\mathcal{W}_{\mathtt{obl}}$. Specifically, under a smoothness and symmetry condition on $\{\mathbf{p}_v, v \in \{-1,1\}^d\}$, which has a parameter $\gamma$ associated with it, we show the following. For $|\mathcal{X}| < \infty$ and $\mathcal{X}_i := \{x(i) : x \in \mathcal{X}\}$, $i \in [d]$, we have

$$\sum_{i=1}^{d} I\big(V(i) \wedge Y^T\big) \leq \frac{C}{2} \cdot T\gamma^2,$$

where the constant $C > 0$ depends only on $\{p_v, v \in \{-1,+1\}^d\}$ and, denoting , is given by $C = (\max_{i \in [d]} |\mathcal{X}_i| - 1) \cdot \max_{x \in \mathcal{X}} \max_{v \in \{-1,+1\}^d} \max_{i \in [d]} \frac{\mathbf{p}_{v \oplus i}(X(i)=x(i))}{\mathbf{p}_v(X(i)=x(i))}$.

**Step 3. Use appropriate difficult instances:** To prove lower bounds for the convex family, we will use the class of functions $\mathcal{G}_{\mathtt{c}} = \{g_v(x) : v \in \{-1,1\}^d\}$ defined on the domain $\mathcal{X} = \{x \in \mathbb{R}^d : \|x\|_\infty \leq b\}$ comprising functions $g_v$ given below:

$$g_v(x) = a \cdot \sum_{i=1}^{d} |x(i) - v(i) \cdot b|, \quad \forall x \in \mathcal{X}, v \in \{-1,1\}^d.$$

On the other hand, to prove lower bounds for the strongly convex family, we will use the class of functions $\mathcal{G}_{\mathtt{sc}} = \{g_v(x) : v \in \{-1,1\}^d\}$ on $\mathcal{X} = \{x \in \mathbb{R}^d : \|x\|_\infty \leq b\}$ given by

$$g_v(x) = a \sum_{i=1}^{d} \Big(\frac{1 + 2\delta v(i)}{2} f_i^+(x) + \frac{1 - 2\delta v(i)}{2} f_i^-(x)\Big), \quad \forall x \in \mathcal{X}, v \in \{-1,1\}^d,$$

where, for $i \in [d]$, $f_i^+(x) = \theta b|x(i)+b| + \frac{1-\theta}{4}(x(i)+b)^2$ and $f_i^-(x) = \theta b|x(i)-b| + \frac{1-\theta}{4}(x(i)-b)^2$.

**Step 4. Carefully combine everything:** We obtain our desired bounds by applying Steps 1 and 2 to difficult instances from Step 3. Since the difficult instance for convex family comprises linear functions, the gradient does not depend on $x$. Thus, we can design oracles which give i.i.d. output with distribution independent of the query point $x_t$, whereby the bound in Step 2 can be applied. Interestingly, we construct different oracles for $p = 1$ and $p \geq 2$.

However, the situation is different for the strongly convex family. The gradients now depend on the query point $x_t$, whereby it is unclear if we can comply with the requirements in Step 2. Interestingly, for communication and local privacy constraints, we construct oracles that allow us to view messages $Y^T$ as output of adaptively selected channels applied to independent samples from a common distribution $\mathbf{p}_v$. While it is unclear if the same can be done for computational constraints as well, we use an alternative approach and exhibit an oracle for which we can find an intermediate message vector $Z_1, \ldots, Z_T$ such that $V$ and $Y^T$ are conditionally independent give $Z^T$ and the message $Z^T$ satisfies the requirements of Step 2.

---

[10]For tuples of random variables $X$ and $Y$, $I(X \wedge Y)$ denotes the mutual information between $X$ and $Y$.

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
