# Information-constrained optimization: can adaptive processing of gradients help?

**Jayadev Acharya**[*]
Cornell University
acharya@cornell.edu

Clément L. Canonne[†]
University of Sydney
clement.canonne@sydney.edu.au

Prathamesh Mayekar[‡]
Indian Institute of Science
prathamesh@iisc.ac.in

Himanshu Tyagi[§]
Indian Institute of Science
htyagi@iisc.ac.in

## Abstract

We revisit first-order optimization under local information constraints such as local privacy, gradient quantization, and computational constraints limiting access to a few coordinates of the gradient. In this setting, the optimization algorithm is not allowed to directly access the complete output of the gradient oracle, but only gets limited information about it subject to the local information constraints.

We study the role of adaptivity in processing the gradient output to obtain this limited information from it. We consider optimization for both convex and strongly convex functions and obtain tight or nearly tight lower bounds for the convergence rate, when adaptive gradient processing is allowed. Prior work was restricted to convex functions and allowed only nonadaptive processing of gradients. For both of these function classes and for the three information constraints mentioned above, our lower bound implies that adaptive processing of gradients cannot outperform nonadaptive processing in most regimes of interest. We complement these results by exhibiting a natural optimization problem under information constraints for which adaptive processing of gradient strictly outperforms nonadaptive processing.

---

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

These lower bounds are seen to match the performance of existing algorithms in most settings, even in settings which were not considered in prior works. For optimization of convex Lipchitz functions over an $\ell_1$ ball using $r$ bits per gradient query, prior work was

restricted to the case $r = O(d)$ only. We show that a simple uniform quantizer used along with repeated queries of the same point is rate-optimal.

The results discussed above show that adaptive processing of gradients does not help for convex optimization over $\ell_p$ balls or even strongly convex optimization over $\ell_2$ balls. This raises the question of whether there are natural function families where adaptive gradient processing can lead to significant savings. Our third contribution is to provide an example of such a family. Specifically, we exhibit a natural optimization problem (entailing $\ell_2$ minimization) under computational constraints for which adaptive gradient processing provides a polynomial factor improvement in convergence rates compared to nonadaptive processing. The key feature of this optimization problem is that the resulting gradients have *structured sparsity*; adaptivity then allows for a two-phase optimization procedure, where the algorithm first "explores" to find the structure before, in a second phase, "exploiting" it to obtain more focused information about the function to minimize. However, nonadaptive gradient processing protocols cannot exploit this hidden structure, as finding it is now akin to locating a needle in a haystack; and thus exhibit much slower convergence rates.

### 1.3 Organization.

The rest of the paper is organized as follows. After formally introducing in Section 2 the setting, the function classes considered (convex and strongly convex), and the information constraints we are concerned with, we state and discuss our lower bounds in Section 3.

In more detail, Section 3.1 focuses on locally differentially private (LDP) optimization, and contains our theorems for convex functions (Theorems 1 and 2 for $p \in [1, 2)$ and $p \in [2, \infty]$, respectively), as well as our lower bound for strongly convex functions (Theorem 3). Section 3.2 contains the analogous results for optimization under communication constraints (Theorems 4 and 5 for convex functions, and Theorem 6 for strongly convex functions). Section 3.3 focuses on optimization with $\ell_2$ loss under computational constraints (*i.e.*, RCD-type schemes), with the lower bound of Theorem 7 for convex functions and that of Theorem 8 for strongly convex functions. Proofs of these lower bounds are given in Section 4.

Finally, Section 5 discusses our example for which adaptive gradient processing does help, with Theorem 12 stating the lower bound for nonadaptive schemes and Theorem 13 providing an upper bound (significantly smaller) for adaptive ones.

**Notation.** Throughout the paper, $q$ denotes the Hölder conjugate of $p$ (that is, $\frac{1}{p} + \frac{1}{q} = 1$). We write $a \vee b$ and $a \wedge b$ for $\max\{a, b\}$ and $\min\{a, b\}$, respectively. We use log for the binary logarithm and ln for the natural logarithm. Information-theoretic quantities, such as mutual information and Kullback–Leibler (KL) divergence, are defined using ln. The iterated logarithm $\ln^*(a)$ is defined as the number of times ln must be iteratively applied to $a$ before the result is at most 1. Finally, we write $\{e_1, \ldots, e_d\}$ for the standard basis of $\mathbb{R}^d$.

## 2 Setup and preliminaries

### 2.1 Optimization under information constraints

We consider the problem of minimizing an unknown convex function $f: \mathcal{X} \to \mathbb{R}$ over its domain $\mathcal{X}$ using *oracle access* to noisy subgradients of the function. That is, the algorithm is not directly given access to the function but can get subgradients of the function at different points of its choice. This class of optimization algorithms includes various descent algorithms, which often provide optimal convergence rate among all the algorithms in this class (*cf.* [23]).

In our setup, gradient estimates supplied by the oracle must pass through a channel $W$,[5] chosen by the algorithm from a fixed set of channels $\mathcal{W}$, and the optimization algorithm $\pi$

---

[5]A channel $W$ with input alphabet $\mathcal{X}$ and output alphabet $\mathcal{Y}$, denoted $W: \mathcal{X} \to \mathcal{Y}$, represents the conditional distribution of the output of a randomized function given its input. In particular, $W(\cdot \mid x)$ is the conditional distribution of the channel given that the input is $x \in \mathcal{X}$.

only has access to the output of this channel. The *channel family* $\mathcal{W}$ represents information constraints imposed in our distributed setting. In detail, the framework is as follows:

1. At iteration $t$, the first-order optimization algorithm $\pi$ makes a query for point $x_t$ to the oracle $O$.

2. Upon receiving the point $x_t$, the oracle outputs $\hat{g}(x_t)$, where $\mathbb{E}\left[\hat{g}(x_t) \mid x_t\right] \in \partial f(x_t)$ and $\partial f(x_t)$ is the subgradient set of function $f$ at $x_t$.

3. The subgradient estimate $\hat{g}(x_t)$ is passed through a channel $W_t \in \mathcal{W}$ and the output $Y_t$ is observed by the first-order optimization algorithm. The algorithm then uses all the messages $\{Y_i\}_{i \in [t]}$ to further update $x_t$ to $x_{t+1}$.

Let $\Pi_T$ be the set of all first-order optimization algorithms that are allowed $T$ queries to the oracle $O$ and after the $t$th query gets back the output $Y_t$ with distribution $W_t(\cdot \mid \hat{g}(x_t))$.

Our goal is to select gradient processing channels $W_t$s and an optimization algorithm $\pi$ to guarantee a small worst-case optimization error. Two classes of *channel selection strategies* are of interest: *adaptive* and *nonadaptive*.

*Adaptive gradient processing.* Under adaptive gradient processing, the channel $W_t$ selected at time $t$ may depend on the previous outputs of channels $\{W_i\}_{i \in [t-1]}$. Specifically, denoting by $Y_t$ the output of the channel used at time $t$, which takes values in the output alphabet $\mathcal{Y}_t$, the *adaptive channel selection strategy* $S := (S_1, \ldots, S_T)$ over $T$ iterations consists of mappings $S_t \colon \mathcal{Y}^{t-1} \to \mathcal{W}$ that take $Y_1, \ldots Y_{t-1}$ as input and output a channel $W_t \in \mathcal{W}$ as output. We write $\mathcal{S}_{\mathcal{W},T}$ for the collection of all such channel selection strategies.

*Nonadaptive gradient processing.* Under nonadaptive selection, all the channels $\{W_t\}_{t \in [T]}$ through which the gradient estimates must pass are decided at the start of the optimization algorithm. In other words, the $W_t$s are independent

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

 \in [1,2)$, $\varepsilon \in [0,1]$, and $D > 0$. There exist absolute constants $c_0, c_1 > 0$ such that, for $T \geq c_0 \frac{d}{\varepsilon^2}$,*

$$\sup_{\mathcal{X}\in\mathbb{X}_p(D)} \mathcal{E}^*(\mathcal{X}, \mathcal{O}_{\mathtt{c},1}, T, \mathcal{W}_{\mathtt{priv},\varepsilon}) \geq \frac{c_1 DB}{\sqrt{T}} \cdot \sqrt{\frac{d}{\varepsilon^2}}.$$

*(Moreover, one can take $c_0 := \frac{1}{2e(e-1)^2}$ and $c_1 := \frac{1}{36(e-1)\sqrt{2e}}$.)*

See Section 4.5 for the proof.

**Theorem 2.** *Let $p \in [2, \infty], \varepsilon \in [0, 1]$, and $D > 0$. There exist absolute constants $c_0, c_1 > 0$ such that, for $T \geq c_0 \frac{d^2}{\varepsilon^2}$,*

$$\sup_{\mathcal{X} \in \mathbb{X}_p(D)} \mathcal{E}^*(\mathcal{X}, \mathcal{O}_{\mathsf{c},p}, T, \mathcal{W}_{\mathtt{priv}, \varepsilon}) \geq \frac{c_1 DB d^{1/2 - 1/p}}{\sqrt{T}} \cdot \sqrt{\frac{d}{\varepsilon^2}}.$$

*(Moreover, one can take $c_0$ and $c_1$ as in Theorem 1.)*

See Section 4.6 for the proof.

**Remark 3** (Tightness of bounds for convex functions and LDP constraints). *[12, Theorem 4 and 5] provide nonadaptive LDP algorithms which show that Theorem 1 is tight up to logarithmic factors for $p = 1$ and Theorem 2 is tight up to constant factors for all $p \in [2, \infty]$ (to the best of our knowledge, no non-trivial upper bound is known for $p \in (1, 2)$.). Therefore, adaptive processing of gradients under LDP cannot significantly improve the convergence rate for convex function families.*

*Interestingly, for $p = 1$, [12] also provide a slightly stronger lower bound of $\frac{c_0 DB}{\sqrt{T}} \cdot \sqrt{\frac{d \log d}{\varepsilon^2}}$ for nonadaptive protocols, which matches the performance of their nonadaptive protocols up to constant factors. This points to a minor gap in our understanding of adaptive protocols: Can we establish a stronger lower bound for adaptive protocols to match the performance of the nonadaptive algorithm of [12], or does there exist a better adaptive protocol? We believe that the latter option is correct, and conjecture that the $\sqrt{d \log d}$ dependence is tight even for adaptive protocols.*

From Remark 1, the standard optimization error for $\ell_1$ and $\ell_p$, $p \in [2, \infty]$, convex family blows up by a factor of $\sqrt{d/\varepsilon^2}$ when the gradient estimates are passed through an $\varepsilon$-LDP channel.

**Strongly convex family.** We prove the following result for strongly convex functions.

**Theorem 3.** *Let $\varepsilon \in [0, 1]$, and $D > 0$. There exist absolute constants $c_0, c_1 > 0$ such that, for $T \geq c_0 \cdot \frac{B^2}{\alpha^2 D^2} \cdot \frac{d}{\varepsilon^2}$,*

$$\sup_{\mathcal{X} \in \mathbb{X}_2(D)} \mathcal{E}^*(\mathcal{X}, \mathcal{O}_{\mathsf{sc}}, T, \mathcal{W}_{\mathtt{priv}, \varepsilon}) \geq \frac{c_1 B^2}{\alpha T} \cdot \frac{d}{\varepsilon^2}.$$

See Section 4.7 for the proof.

**Remark 4** (Tightness of bounds for strongly convex functions and LDP constraints). *One can use stochastic gradient descent with the nonadaptive protocol from [12, Appendix C.2] to obtain a nonadaptive protocol with convergence rate matching the lower bound in Theorem 3 up to constant factors, establishing that adaptivity does not help for strongly convex functions.*

From Remark 2, the standard optimization error for strongly convex functions blows up by a factor of $\frac{d}{\varepsilon^2}$ when the gradient estimates are passed through an $\varepsilon$-LDP channel.

### 3.2 Lower bounds on communication-constrained optimization

**Convex function family.** For convex functions, we prove the following lower bounds.

**Theorem 4.** *Let $p \in [1, 2)$, and $D > 0$. There exists an absolute constant $c_0 > 0$ such that, for $r \in \mathbb{N}$, and $T \geq \frac{d}{6r}$,*

$$\sup_{\mathcal{X} \in \mathbb{X}_p(D)} \mathcal{E}^*(\mathcal{X}, \mathcal{O}_{\mathsf{c},1}, T, \mathcal{W}_{\mathtt{com}, r}) \geq \frac{c_0 DB}{\sqrt{T}} \cdot \sqrt{\frac{d}{d \wedge r}}.$$

*(Moreover, one can take $c_0 := \frac{1}{12\sqrt{58}}$.)*

See Section 4.7 for the proof.

**Theorem 5.** *Let $p \in [2, \infty]$, and $D > 0$. There exists an absolute constant $c_0 > 0$ such that, for $r \in \mathbb{N}$, and $T \geq \frac{1}{4} \cdot \frac{d^2}{2^r \wedge d}$, we have*

$$\sup_{\mathcal{X} \in \mathbb{X}_p(D)} \mathcal{E}^*(\mathcal{X}, \mathcal{O}_{\mathsf{c},p}, T, \mathcal{W}_{\mathsf{com},r}) \geq \left( \frac{c_0 D B d^{1/2 - 1/p}}{\sqrt{T}} \cdot \sqrt{\frac{d}{d \wedge 2^r}} \right) \vee \left( \frac{c_0 D B}{\sqrt{T}} \cdot \sqrt{\frac{d}{d \wedge r}} \right)$$

*(Moreover, one can take $c_0 := \frac{1}{12\sqrt{58}}$.)*

See Section 4.6 for the proof.

**Remark 5** (Tightness of bounds for convex functions and communication constraints). *In Appendix B, we provide a scheme which matches the lower bound in Theorem 4 for $p = 1$ up to constant factors for any $r$. Since each coordinate of oracle output is bounded by $B$ for $p = 1$, we simply can use an unbiased 1-bit quantizer for each coordinate. The proposed scheme uses such a quantizer for each coordinate and makes $d/r$ repeated queries to the oracle for the same point, but gets 1-bit information about $r$ different coordinates in each query.*

*In general, there are two obstacles in extending this scheme to other cases: First, the uniform bound of $B$ for each coordinate is too loose. Second, we cannot assume that repeated queries for the same point give identically distributed outputs (we only assume that their means are subgradients and they have bounded moments). We were able to circumvent the second difficulty for $p = 1$ using convexity of the set of subgradients. However, in general, it remains an obstacle. Nonetheless, if we make the assumption that repeated queries yield i.i.d. outputs, we can even attain the lower bound Theorem 5 for $p = \infty$ up to a constant factor as follows. We can use the quantizer SimQ from [20] to obtain an unbiased estimator of the common mean (a subgradient) of the repeated query outputs, which takes only $d$ distinct values. We can then apply the simulate-and-infer approach from [3] to obtain samples from this $d$-ary distribution using $r$ bits per query and $O(d/2^r)$ queries per sample. This results in an $O(d/r)$ factor blow-up in the standard convergence rate, which when used with appropriate mirror descent algorithms matches our lower bound in Theorem 5 for $p = \infty$.*

*In general, without making any additional assumptions about the oracle, we can use the quantizer SimQ$^+$ from [20] with $k = r$ and appropriate mirror descent algorithms to get upper bounds that match the lower bounds in Theorem 5 for $p \in [2, \infty]$, up to an additional $O(\log d)$ factor. For $p = 2$, we can use the quantizer RATQ from [21] to improve this match to an $O(\ln \ln^* d)$ factor. However, as was the case in the privacy setting, to the best of our knowledge no non-trivial upper bound is known for $p \in (1, 2)$.*

From Remark 1, the standard optimization errors for $\ell_1$ and $\ell_p$, $p \in [2, \infty]$, convex family blow up by a factor of $\sqrt{\frac{d}{d \wedge r}}$ and $\sqrt{\frac{d}{d \wedge 2^r}} \vee \sqrt{\frac{d^{2/p}}{d \wedge r}}$, respectively, when the gradient estimates are compressed to $r$ bits.

**Strongly convex family.**  We prove the following result for strongly convex functions.

**Theorem 6.** *Let $D > 0$. There exist absolute constants $c_0, c_1 > 0$ such that, for $r \in \mathbb{N}$ and $T \geq c_0 \cdot \frac{B^2}{\alpha^2 D^2} \cdot \frac{d}{r}$,*

$$\sup_{\mathcal{X} \in \mathbb{X}_2(D)} \mathcal{E}^*(\mathcal{X}, \mathcal{O}_{\mathsf{sc}}, T, \mathcal{W}_{\mathsf{com},r}) \geq \frac{c_1 B^2}{\alpha T} \cdot \frac{d}{d \wedge r} .$$

See Section 4.7 for the proof.

**Remark 6** (Tightness of bounds for strongly convex functions and communication constraints). *We note that the nonadaptive scheme RATQ in [21] along with stochastic gradient descent matches the lower bound in Theorem 6 up to a $\ln \ln^* d$ factor for $r = \Omega(\ln \ln^* d)$.*

From Remark 2, the standard optimization error for strongly convex functions blows up by a factor of $\frac{d}{r}$ when the gradient estimates are compressed to $r$ bits.

### 3.3 Lower bounds on computationally-constrained optimization

We restrict to the case of Euclidean geometry ($p = 2$) for the oblivious sampling channel family $\mathcal{W}_{\mathtt{obl}}$. Our motivation for introducing this class was to study the optimality of standard RCD, which is proposed to work in the Euclidean setting alone. Furthermore, if we consider a slightly larger family of channels where the sampling probabilities can depend on the input itself, the resulting family will be similar to the 1-bit communication family, which we have addressed in Section 3.2.

**Convex family.** For convex functions, we establish the following lower bound, for $p = 2$.

**Theorem 7.** *Let $D > 0$. There exists an absolute constant $c_0 > 0$ such that, for $T \geq \frac{d}{4}$, we have*

$$\sup_{\mathcal{X} \in \mathbb{X}_2(D)} \mathcal{E}^*(\mathcal{X}, \mathcal{O}_{\mathtt{c},2}, T, \mathcal{W}_{\mathtt{obl}}) \geq \frac{c_0 \sqrt{d} D B}{\sqrt{T}} \, .$$

*(Moreover, one can take $c_0 := \frac{1}{72}$.)*

See Section 4.5 for a proof.

The standard Random Coordinate Descent (RCD) (see for instance [10, Theorem 6.6]), which employs uniform sampling, matches this lower bound up to constant factors. The optimality of standard RCD motivates further the folklore approach of uniformly sampling coordinates for random coordinate descent unless there is an obvious structure to exploit (as in [24]). This establishes that adaptive sampling strategies do not improve over nonadaptive sampling strategies for the family $\mathcal{W}_{\mathtt{obl}}$. Also from Remark 1, the standard optimization error for $\ell_2$ convex family blows up by a factor of $\sqrt{d}$ when the gradients are sampled obliviously.

**Strongly convex family.** For strongly convex functions, we obtain the following lower bound, for $p = 2$.

**Theorem 8.** *Let $D > 0$. There exist absolute constants $c_0, c_1 > 0$ such that, for $T \geq c_0 \cdot d \frac{B^2}{\alpha^2 D^2}$, we have*

$$\sup_{\mathcal{X} \in \mathbb{X}_2(D)} \mathcal{E}^*(\mathcal{X}, \mathcal{O}_{\mathtt{sc}}, T, \mathcal{W}_{\mathtt{obl}}) \geq \frac{c_1 d B^2}{\alpha T} \, .$$

See Section 4.7 for the proof.

Once again, the standard RCD algorithm matches this lower bound, which shows that adaptive sampling strategies do not improve over nonadaptive sampling strategies for strongly convex optimization. Further, from Remark 2, the standard optimization error for strongly convex family blows up by a factor of $d$ when the gradients are sampled obliviously.

## 4 Proofs of average information lower bounds

### 4.1 Outline of the proof for our lower bounds

The proofs of our lower bounds for adaptive protocols follow the same general template, summarized below.

**Step 1. Relating optimality gap to average information:** We consider a family of functions $\mathcal{G} = \{g_v : v \in \{-1, 1\}^d\}$ satisfying suitable conditions and associate with it a "discrepancy metric" $\psi(\mathcal{G})$ that allows us to relate the optimality gap of any algorithm to an average mutual information quantity. Specifically, for $V$ distributed uniformly over $\{-1, 1\}^d$, we show that the output $\hat{x}$ of any optimization algorithm satisfies

$$\mathbb{E}\left[g_V(\hat{x}) - \min_{x \in \mathcal{X}} g_V(x)\right] \geq \frac{d\psi(\mathcal{G})}{6}\left[1 - \sqrt{\frac{2}{d}\sum_{i=1}^{d} I(V(i) \wedge Y^T)}\right],$$

where $Y_t$ is the channel output for the gradient in the $t$th iteration and $Y^T := (Y_1, \ldots, Y_T)$.

Heuristically, we have related the gap to optimality to the difficulty of inferring $V$ by observing $Y^T$. We note that the bound above is similar to that of [5], but instead of mutual information $I(V \wedge Y^T)$ we get the average mutual information per coordinate. This latter quantity is amenable to analysis for adaptive protocols.

**Step 2. Average information bounds:** To bound the average mutual information per coordinate, $\frac{1}{d} \sum_{i=1}^{d} I(V(i) \wedge Y^T)$, we take recourse to the recently proposed bounds from [2]. These bounds hold for $Y^T$ which is the output of adaptively selected channels from a fixed channel family $\mathcal{W}$, with i.i.d. input $X^T = (X_1, \ldots, X_T)$ generated from a family of distributions $\{\mathbf{p}_v, v \in \{-1,1\}^d\}$. We view the output of oracle as inputs $X^T$ and derive the required bound.

While results in [2] provided bounds for $\mathcal{W}_{\mathtt{priv},\varepsilon}$ and $\mathcal{W}_{\mathtt{comm},r}$, we extend the approach to handle $\mathcal{W}_{\mathtt{obl}}$. Specifically, under a smoothness and symmetry condition on $\{\mathbf{p}_v, v \in \{-1,1\}^d\}$, which has a parameter $\gamma$ associated with it, we show the following:

For $|\mathcal{X}| < \infty$ and $\mathcal{X}_i := \{x(i) : x \in \mathcal{X}\}$, $i \in [d]$, we have

$$\sum_{i=1}^{d} I(V(i) \wedge Y^T) \leq \frac{C}{2} \cdot T\gamma^2,$$

where the constant $C$ depends only on $\{\mathbf{p}_v, v \in \{-1,+1\}^d\}$ and, denoting by $v^{\oplus i} \in \{-1,1\}^d$ the vector with the sign of the $i$th coordinate of $v$ flipped, is given by

$$C = (\max_{i \in [d]} |\mathcal{X}_i| - 1) \cdot \max_{x \in \mathcal{X}} \max_{v \in \{-1,+1\}^d} \max_{i \in [d]} \frac{\mathbf{p}_{v^{\oplus i}}(X(i) = x(i))}{\mathbf{p}_v(X(i) = x(i))}.$$

**Step 3. Use appropriate difficult instances** On the one hand, to prove lower bounds for the convex family we will use the class of functions $\mathcal{G}_{\mathtt{c}} = \{g_v(x) : v \in \{-1,1\}^d\}$ defined on the domain $\mathcal{X} = \{x \in \mathbb{R}^d : \|x\|_\infty \leq b\}$ comprising functions $g_v$ given below:

$$g_v(x) = a \cdot \sum_{i=1}^{d} |x(i) - v(i) \cdot b|, \quad \forall x \in \mathcal{X}, v \in \{-1,1\}^d.$$

On the other hand, to prove lower bounds for the strongly convex family, we will use the class of functions $\mathcal{G}_{\mathtt{sc}} = \{g_v(x) : v \in \{-1,1\}^d\}$ on $\mathcal{X} = \{x \in \mathbb{R}^d : \|x\|_\infty \leq b\}$ given by

$$g_v(x) = a \sum_{i=1}^{d} \left( \frac{1 + 2\delta v(i)}{2} f_i^+(x) + \frac{1 - 2\delta v(i)}{2} f_i^-(x) \right), \quad \forall x \in \mathcal{X}, v \in \{-1,1\}^d,$$

where $f_i^+$ and $f_i^-$, for $i \in [d]$, are given by

$$f_i^+(x) = \theta b |x(i) + b| + \frac{1-\theta}{4}(x(i) + b)^2, \qquad f_i^-(x) = \theta b |x(i) - b| + \frac{1-\theta}{4}(x(i) - b)^2.$$

**Step 4. Carefully combine everything:** We obtain our desired bounds by applying Steps 1 and 2 to difficult instances from Step 3. Since the difficult instance for convex family consists of linear functions, the gradient does not depend on $x$. Thus, we can design oracles which give i.i.d. output with distribution independent of the query point $x_t$, whereby the bound in Step 2 can be applied. Interestingly, we construct different oracles for $p < 2$ and $p \geq 2$.

However, the situation is different for the strongly convex family. The gradients now depend on the query point $x_t$, whereby it is unclear if we can comply with the requirements in Step 2. Interestingly, for communication and local privacy constraints, we construct oracles that allow us to view messages $Y^T$ as the output of adaptively selected channels applied to independent samples from a common distribution $\mathbf{p}_v$. While it is unclear if the same can be done for computational constraints as well, we use an alternative approach and exhibit an oracle for which we can find an intermediate message vector $Z_1, \ldots, Z_T$ such that (i) $V$ and $Y^T$ are conditionally independent given $Z^T$ and (ii) the message $Z^T$ satisfies the requirements of Step 2.

## 4.2 Relating optimality gap to average information

In this section, we prove a general lower bound for the expected gap to optimality by considering a parameterized family of functions and oracles which is contained in our oracle family of interest. We present a bound that relates the expected gap to optimality to the average mutual information between the channel output and different coordinates of the unknown parameter. This step is the key difference between our approach and that of [5], which used Fano's method instead of our bound below. We remark that the bounds resulting from Fano's method are typically not amenable to analysis for adaptive protocols.

In more detail, our result can be used to prove bounds for the average optimization error over any class of functions which satisfies the two conditions below.

**Assumption 1.** *Let $\mathcal{X} \subseteq \mathbb{R}^d$ and $\mathcal{V} = \{-1, 1\}^d$. Let $\mathcal{G} = \{g_v : v \in \mathcal{V}\}$ where $g_v : \mathcal{X} \to \mathbb{R}$ are real-valued functions from $\mathcal{X}$ such that*

1. *the $g_v$s are* coordinate-wise decomposable, *i.e., there exist functions $g_{i,b} \colon \mathbb{R} \to \mathbb{R}$, $i \in [d]$, $b \in \{-1, 1\}$, such that*

$$g_v(x) = \sum_{i=1}^{d} g_{i,v(i)}(x(i)).$$

2. *the minimum of $g_v$ is also a* coordinate-wise minimum, *i.e., if we denote by $x_v^*$ the minimum of $g_v$ over $\mathcal{X}$, then, for all $i \in [d]$, we have*

$$x_v^*(i) = \arg\min_{y \in \mathcal{X}_i} g_{i,v(i)}(y),$$

*where $\mathcal{X}_i = \{x(i) : x \in \mathcal{X}\}$.*

For $\mathcal{G}$ satisfying Assumptions 1 and for $i \in [d]$, we now define the following discrepancy metric:

$$\psi_i(\mathcal{G}) := \min_{y \in \mathcal{X}_i} \left( g_{i,1}(y) + g_{i,-1}(y) - \left( \min_{y' \in \mathcal{X}_i} g_{i,1}(y') + \min_{y' \in \mathcal{X}_i} g_{i,-1}(y') \right) \right) \tag{8}$$

$$\psi(\mathcal{G}) := \min_{i \in [d]} \psi_i(\mathcal{G}). \tag{9}$$

This is a "coordinate-wise counterpart" of the metric used in [5]. The next lemma follows readily from this definition.

**Lemma 2.** *Fix $i \in [d]$. For every $y \in \mathcal{X}_i$, there can be at most one $b \in \{-1, 1\}$ such that*

$$g_{i,b}(y) - \min_{y' \in \mathcal{X}_i} g_{i,b}(y') \leq \frac{\psi_i(\mathcal{G})}{3}.$$

*Proof.* Let $b \in \{-1, 1\}$. By definition of $\psi_i(\mathcal{G})$, for all $y \in \mathcal{X}_i$ we have

$$\left( g_{i,b}(y) - \min_{y' \in \mathcal{X}_i} g_{i,b}(y') \right) + \left( g_{i,-b}(y) - \min_{y' \in \mathcal{X}_i} g_{i,-b}(y') \right) \geq \psi_i(\mathcal{G}).$$

For $y$ such that $g_{i,b}(y) - \min_{y' \in \mathcal{X}_i} g_{i,b}(y') \leq \frac{\psi_i(\mathcal{G})}{3}$, we now must have that

$$g_{i,-b}(y) - \min_{y' \in \mathcal{X}_i} g_{i,-b}(y') \geq \frac{2\psi_i(\mathcal{G})}{3}. \qquad \square$$

We will use this observation to bound the expected gap to optimality for any algorithm $\pi$ optimizing an unknown function in $\mathcal{G}$ that has access to only the corresponding first-order oracle.

**Lemma 3.** *Suppose $\mathcal{G} = \{g_v : v \in \{-1, 1\}^d\}$ satisfies Assumption 1. Let $\pi$ be any optimization algorithm that adaptively selects the channels $\{W_j\}_{j \in [T]}$. For a random variable $V$*

*distributed uniformly over $\{-1, 1\}^d$, the output $\hat{x}$ of $\pi$ when it is applied to a function from $\mathcal{G}$ and any associated (stochastic subgradient) oracle satisfies*

$$\mathbb{E}\left[g_V(\hat{x}) - g_V(x_V^*)\right] \geq \frac{d\psi(\mathcal{G})}{6}\left[1 - \sqrt{\frac{1}{d}\sum_{i=1}^{d} 2I(V(i) \wedge Y^T)}\right],$$

*where $\psi(\mathcal{G}) = \min_{j \in [d]} \psi_j(\mathcal{G})$, $Y_t$ is the channel output for the gradient at time step $t$ and $Y^T := (Y_1, \ldots, Y_T)$.*

*Proof.* Our proof is based on relating the gap to optimality to the error in estimation of $V$ upon observing $Y^T$. Suppose the algorithm $\pi$ along with channels $\{W_j\}_{j \in [T]}$ outputs the point $\hat{x}$ after $T$ iterations. By linearity of expectation, the decomposability of $g_v$, and Markov's inequality, we have

$$\mathbb{E}\left[g_V(\hat{x}) - g_V(x_V^*)\right] = \sum_{i=1}^{d} \mathbb{E}\left[g_{i, V(i)}(\hat{x}(i)) - g_{i, V(i)}(x_V^*(i))\right]$$

$$\geq \sum_{i=1}^{d} \frac{\psi_i(\mathcal{G})}{3} \Pr\left(g_{i, V(i)}(\hat{x}(i)) - g_{i, V(i)}(x_V^*(i)) \geq \frac{\psi_i(\mathcal{G})}{3}\right)$$

$$\geq \frac{\psi(\mathcal{G})}{3} \sum_{i=1}^{d} \Pr\left(g_{i, V(i)}(\hat{x}(i)) - g_{i, V(i)}(x_V^*(i)) \geq \frac{\psi_i(\mathcal{G})}{3}\right). \tag{10}$$

We proceed to bound each summand separately.

Fix any $i \in [d]$ and consider the following estimate for $V(i)$: Given $\hat{x}$, we output a $\hat{V}(i) \in \{-1, 1\}$ satisfying

$$g_{i, \hat{V}(i)}(\hat{x}(i)) - \min_{y' \in \mathcal{X}_i} g_{i, \hat{V}(i)}(y') < \frac{\psi_i(G)}{3};$$

if no such $\hat{V}(i)$ exists, we generate $\hat{V}(i)$ uniformly from $\{-1, 1\}$. Then, as a consequence of Lemma 2, we get

$$\Pr\left(\hat{V}(i) \neq v(i)\right) \leq \Pr\left(g_{i, v(i)}(\hat{x}(i)) - g_{i, v(i)}(x_v^*(i)) \geq \frac{\psi_i(\mathcal{G})}{3}\right). \tag{11}$$

Next, denote by $\mathbf{p}^{Y^T}$ the distribution of $Y^T$ and by $\mathbf{p}_{+i}^{Y^T}$ and $\mathbf{p}_{-i}^{Y^T}$, respectively, the distributions of $Y^T$ given $V(i) = +1$ and $V(i) = -1$. It is easy to verify that

$$\mathbf{p}^{Y^T} = \frac{1}{2}(\mathbf{p}_{+i}^{Y^T} + \mathbf{p}_{-i}^{Y^T}), \quad \forall i \in [d].$$

Noting that $V(i)$ is uniform and the estimate $\hat{V}(i)$ is formed as a function of $Y^T$, we get

$$\Pr\left(\hat{V}(i) \neq v(i)\right) \geq \frac{1}{2} - \frac{1}{2}d_{\text{TV}}\left(\mathbf{p}_{+i}^{Y^T}, \mathbf{p}_{-i}^{Y^T}\right). \tag{12}$$

From this, combining (11) and (12) and plugging the result into (10), we have

$$\mathbb{E}\left[g_v(\hat{x}) - g_v(x_v^*)\right] \geq \frac{\psi(\mathcal{G})}{6} \sum_{i=1}^{d} \left[1 - d_{\text{TV}}\left(\mathbf{p}_{+i}^{Y^T}, \mathbf{p}_{-i}^{Y^T}\right)\right]$$

$$\geq \frac{\psi(\mathcal{G})}{6} \sum_{i=1}^{d} \left[1 - d_{\text{TV}}\left(\mathbf{p}_{+i}^{Y^T}, \mathbf{p}^{Y^T}\right) - d_{\text{TV}}\left(\mathbf{p}_{-i}^{Y^T}, \mathbf{p}^{Y^T}\right)\right]$$

$$\geq \frac{\psi(\mathcal{G})}{6} \sum_{i=1}^{d} \left[1 - \sqrt{\frac{1}{2}D(\mathbf{p}_{+i}^{Y^T} \| \mathbf{p}^{Y^T})} - \sqrt{\frac{1}{2}D(\mathbf{p}_{-i}^{Y^T} \| \mathbf{p}^{Y^T})}|\right]$$

$$\geq \frac{d\psi(\mathcal{G})}{6} \left[1 - \sqrt{\frac{1}{d}\sum_{i=1}^{d}D(\mathbf{p}_{+i}^{Y^T} \| \mathbf{p}^{Y^T}) + D(\mathbf{p}_{-i}^{Y^T} \| \mathbf{p}^{Y^T})}\right]$$

$$= \frac{d\psi(\mathcal{G})}{6} \left[ 1 - \sqrt{\frac{2}{d} \sum_{i=1}^{d} I(V(i) \wedge Y^T)} \right],$$

where the second inequality follows from the triangle inequality, the third is Pinsker's inequality, and the fourth is Jensen's inequality. □

## 4.3 Average information bounds

The next step in our proof is to bound the average mutual information that emerged in Section 4.2. A general recipe for bounding this average mutual information has been given recently in [2], which we recall below.

Let $\{\mathbf{p}_v, v \in \{-1,1\}^d\}$ be a family of distributions over some domain $\mathcal{X}$ and $\mathcal{W}$ be a fixed channel family. For $v \in \{-1,1\}^d$ and $i \in [d]$, denote by $v^{\oplus i}$ the element of $\{-1,1\}^d$ obtained by flipping the $i$th coordinate of $v$. For a fixed $v$, we obtain $T$ independent samples $X_1, \ldots, X_T$ from $\mathbf{p}_v$. Let $Y_1, \ldots, Y_T$ be the output of channels selected from the channel family $\mathcal{W}$ by an adaptive channel selection strategy (see Section 2.1) when input to the channel at time $t$ is $X_t$, $1 \leq t \leq T$.[7]

For $V$ distributed uniformly on $\{-1,1\}^d$, we are interested in bounding $(1/d) \sum_{i=1}^{d} I(V(i) \wedge Y^T)$. In [2], different bounds were given for this quantity under different assumptions. We state these assumptions below.

**Assumption 2.** *For every $v \in \{-1,1\}^d$ and $i \in [d]$, there exists $\phi_{v,i} \colon \mathcal{X} \to \mathbb{R}$ such that $\mathbb{E}_{\mathbf{p}_v}\left[\phi_{v,i}^2\right] = 1$, $\mathbb{E}_{\mathbf{p}_v}\left[\phi_{v,i}\phi_{v,j}\right] = \mathbb{1}_{\{i=j\}}$ holds for all $i, j \in [d]$, and*

$$\frac{d\mathbf{p}_{v^{\oplus i}}}{d\mathbf{p}_v} = 1 + \gamma\phi_{v,i},$$

*where $\gamma \in \mathbb{R}$ is a fixed constant independent of $v, i$.*

**Assumption 3.** *There exists some $\kappa_{\mathcal{W}} \geq 1$ such that*

$$\max_{v \in \{-1,1\}^d} \max_{y \in \mathcal{Y}} \sup_{W \in \mathcal{W}} \frac{\mathbb{E}_{\mathbf{p}_{v^{\oplus i}}}\left[W(y \mid X)\right]}{\mathbb{E}_{\mathbf{p}_v}\left[W(y \mid X)\right]} \leq \kappa_{\mathcal{W}}.$$

**Assumption 4.** *There exists some $\sigma \geq 0$ such that, for all $v \in \{-1,1\}^d$, the vector $\phi_v(X) := (\phi_{v,i}(X))_{i \in [d]} \in \mathbb{R}^d$ is $\sigma^2$-subgaussian for $X \sim \mathbf{p}_v$.[8] Further, for any fixed $z$, the random variables $\phi_{v,i}(X)$ are independent across $i \in [d]$.*

We then have the following bound local privacy constraints.

**Theorem 9** ([2, Corollary 6]). *Consider $\{\mathbf{p}_v, v \in \{-1,1\}^d\}$ satisfying Assumption 2 and the channel family $\mathcal{W} = \mathcal{W}_{\mathtt{priv},\varepsilon}$. Let $V$ be distributed uniformly over $\{-1,1\}^d$ and $Y^T$ be the output of channels selected by the optimization algorithm as above. Then, we have*

$$\sum_{i=1}^{d} I(V(i) \wedge Y^T) \leq T \cdot \frac{\gamma^2}{2} \cdot e^\varepsilon (e^\varepsilon - 1)^2.$$

For the case of communication constraints, we have the analogous statement below:

**Theorem 10** ([2, Corollary 6]). *Consider $\{\mathbf{p}_v, v \in \{-1,1\}^d\}$ satisfying Assumptions 2 and 3 and the channel family $\mathcal{W} = \mathcal{W}_{\mathtt{com},r}$. Let $V$ be distributed uniformly over $\{-1,1\}^d$ and $Y^T$ be the output of channels selected by the optimization algorithm as above. Then, we have*

$$\sum_{i=1}^{d} I(V(i) \wedge Y^T) \leq \frac{1}{2}\kappa_{\mathcal{W}_{\mathtt{com},r}} \cdot T\gamma^2 (2^r \wedge d).$$

---

[7]The bound in [2] allows even shared randomness $U$ in its definition of interactive protocols. We have omitted $U$ in this paper for simplicity.

[8]Recall that a random variable $Y$ is $\sigma^2$-subgaussian if $\mathbb{E}[Y] = 0$ and $\mathbb{E}\left[e^{\lambda Y}\right] \leq e^{\sigma^2\lambda^2/2}$ for all $\lambda \in \mathbb{R}$; and that a vector-valued random variable $Y$ is $\sigma^2$-subgaussian if its projection $\langle Y, u \rangle$ is $\sigma^2$-subgaussian for every unit vector $u$.

*Moreover, if Assumption 4 holds as well, we have*

$$\sum_{i=1}^{d} I\big(V(i) \wedge Y^T\big) \leq (\ln 2)\kappa_{\mathcal{W}_{\mathrm{com}, r}} \sigma^2 \cdot T\gamma^2 r.$$

Finally, we derive a bound for the oblivious sampling channel family.

**Theorem 11.** *Consider $\{\mathbf{p}_v, v \in \{-1, 1\}^d\}$ satisfying Assumption 2 and the channel family $\mathcal{W} = \mathcal{W}_{\mathrm{obl}}$. Let $V$ be distributed uniformly over $\{-1, 1\}^d$ and $Y^T$ be the output of channels selected by the optimization algorithm as above. Further, assume that $|\mathcal{X}| < \infty$. Then, we have*

$$\sum_{i=1}^{d} I\big(V(i) \wedge Y^T\big) \leq \frac{C}{2} \cdot T\gamma^2,$$

*where the constant $C$ depends only on $\{\mathbf{p}_v, v \in \{-1, +1\}^d\}$ and, denoting $\mathcal{X}_i := \{x(i) : x \in \mathcal{X}\}$, is given by*

$$C = (\max_{i \in [d]} |\mathcal{X}_i| - 1) \cdot \max_{x \in \mathcal{X}} \max_{v \in \{-1, +1\}^d} \max_{i \in [d]} \frac{\mathbf{p}_{v \oplus i}(X(i) = x(i))}{\mathbf{p}_v(X(i) = x(i))}.$$

*Proof.* We recall another result from [2, Theorem 5]: Under Assumptions 2 and 3, we have[9]

$$\sum_{i=1}^{d} I\big(V(i) \wedge Y^T\big) \leq \frac{1}{2}\kappa_{\mathcal{W}_{\mathrm{obl}}} \cdot T\gamma^2 \max_{v \in \{-1, 1\}^d} \max_{W \in \mathcal{W}_{\mathrm{obl}}} \sum_{y \in \mathcal{Y}} \frac{\mathrm{Var}_{\mathbf{p}_v}[W(y \mid X)]}{\mathbb{E}_{\mathbf{p}_v}[W(y \mid X)]}.$$

We now evaluate various parameters involved in this bound. Let $W$ be a oblivious sampling channel specified by the probability vector $(p_i)_{i \in [d]}$. Note that a channel $W \in \mathcal{W}_{\mathrm{obl}}$ can be equivalently viewed as having output alphabet $\mathcal{Y} = \{(i, z) : z \in \mathcal{X}_i, i \in [d]\}$. Recall that for an input $x$, the channel output is $x(i)$ with probability $p_i$, $i \in [d]$, i.e., for $y = (i, z)$, $W(y \mid x) = p_i \mathbb{1}_{\{x(i) = z\}}$. Thus, we have

$$\sum_{y \in \mathcal{Y}} \frac{\mathrm{Var}_{\mathbf{p}_v}[W(y \mid X)]}{\mathbb{E}_{\mathbf{p}_v}[W(y \mid X)]} = \sum_{i=1}^{d} \sum_{z \in \mathcal{X}_i} \frac{p_i^2 \Pr(X(i) = z) - p_i^2 \Pr(X(i) = z)^2}{p_i \Pr(X(i) = z)}$$

$$= \sum_{i=1}^{d} p_i(|\mathcal{X}_i| - 1)$$

$$\leq \max_{i \in [d]} |\mathcal{X}_i| - 1.$$

Furthermore, proceeding similarly, we get that Assumption 3 holds as well with

$$\kappa_{\mathcal{W}_{\mathrm{obl}}} = \max_{x \in \mathcal{X}} \max_{v \in \{-1, +1\}^d} \max_{i \in [d]} \frac{\mathbf{p}_{v \oplus i}(X(i) = x(i))}{\mathbf{p}_v(X(i) = x(i))}.$$

The proof is completed by combining the bounds above. $\qquad\square$

### 4.4 The difficult instances for our lower bounds

With our general tools ready, we now describe the precise constructions of function families we use to get our lower bounds. We first provide the details of a family $\mathcal{G}_{\mathtt{c}}(a, b)$ of convex functions, before turning to $\mathcal{G}_{\mathtt{sc}}(a, b, \delta, \theta)$, our family of hard instances for the strongly convex setting. In both cases, our families of hard instances are parameterized (by $a, b$ and $a, b, \delta, \theta$, respectively), and setting those parameters carefully will enable us to prove our various results.

---

[9]This is the general bound underlying Theorem 9.

**Difficult functions for the convex family.** To prove lower bounds for the convex family, we will use the class of functions $\mathcal{G}_c(a,b)$ below, parameterized by $a,b > 0$ and defined on the domain $\mathcal{X}$ as follows:

$$\mathcal{X} = \{x \in \mathbb{R}^d : \|x\|_\infty \leq b\},$$

$$g_v(x) = a \cdot \sum_{i=1}^d |x(i) - v(i) \cdot b|, \quad \forall x \in \mathcal{X}, v \in \{-1,1\}^d, \text{ and}$$

$$\mathcal{G}_c = \{g_v(x) : v \in \{-1,1\}^d\}. \tag{13}$$

Observe that the class $\mathcal{G}_c$ satisfies the conditions in Assumption 1 with $g_{i,1}(x) = a|x(i) - b|$ and $g_{i,-1}(x) = a|x(i) + b|$ and $\mathcal{X}_i = [-b,b]$ for all $i \in [d]$. Further, we can bound the discreprency metric for this class as follows.

**Lemma 4.** *For the class of functions $\mathcal{G}_c$ defined in* (13), *we have* $\psi(\mathcal{G}_c) \geq 2ab$.

*Proof.* Note that $\min_{x \in [-b,b]} g_{i,1}(x) = \min_{x \in [-b,b]} g_{i,-1}(x) = 0$. Therefore, for all $i \in [d]$,

$$\psi_i(\mathcal{G}_c) = \min_{x \in [-b,b]} (a|x(i) - b| + a|x(i) + b|) \geq 2ab,$$

where the inequality follows from the triangle inequality. $\square$

**Difficult functions for the strongly convex family.** To prove lower bounds for the strongly convex family, we will use the class of functions $\mathcal{G}_{sc}(a,b,\delta,\theta)$, parameterized by $a, b > 0$, $\delta > 0$, and $\theta \in [0,1]$, and defined on the domain $\mathcal{X}$ as follows:

$$\mathcal{X} = \{x \in \mathbb{R}^d : \|x\|_\infty \leq b\},$$

$$g_v(x) = a \sum_{i=1}^d \left( \frac{1 + 2\delta v(i)}{2} f_i^+(x) + \frac{1 - 2\delta v(i)}{2} f_i^-(x) \right), \quad \forall x \in \mathcal{X}, v \in \{-1,1\}^d, \text{ and}$$

$$\mathcal{G}_{sc} = \{g_v(x) : v \in \{-1,1\}^d\}, \tag{14}$$

where $f_i^+$ and $f_i^-$, for $i \in [d]$, are given by

$$f_i^+(x) = \theta b |x(i) + b| + \frac{1-\theta}{4}(x(i) + b)^2, \tag{15}$$

$$f_i^-(x) = \theta b |x(i) - b| + \frac{1-\theta}{4}(x(i) - b)^2, \tag{16}$$

for all $x \in \mathcal{X}$. We can check that, for every $v \in \{-1,1\}^d$, the function $g_v$ is then $\alpha$-strongly convex for $\alpha := a \cdot \frac{1-\theta}{4}$. Moreover, we have the following bound for the discrepancy metric.

**Lemma 5.** *For the class of functions $\mathcal{G}_{sc}$ defined in* (14), *if $\frac{1-\theta}{1+\theta} \geq 2\delta$ then $\psi(\mathcal{G}_{sc}) \geq \frac{2ab^2\delta^2}{1-\theta}$.*

*Proof.* This follows from similar calculations as in [5, Appendix A]; we provide the proof here for completeness. Fixing any $v \in \{-1,1\}^d$, we first note that by definition of $\mathcal{G}_{sc}$, the function $g_v$ can be indeed be decomposed as $g_v(x) = \sum_{i=1}^d g_{i,v(i)}(x_i)$ for $x \in \mathcal{X}$ (i.e., $\|x\|_\infty \leq b$), where, for $i \in [d]$, $\nu \in \{-1,1\}$ and $y \in \mathcal{X}_i := [-b,b]$,

$$g_{i,\nu}(y) = a \left( \frac{1 + 2\delta\nu}{2} \left( \theta b |y + b| + \frac{1-\theta}{4}(y+b)^2 \right) + \frac{1 - 2\delta\nu}{2} \left( \theta b |y - b| + \frac{1-\theta}{4}(y-b)^2 \right) \right)$$

$$= a \left( \frac{1-\theta}{4} y^2 + \frac{1+3\theta}{4} b^2 + \delta\nu(1+\theta)by \right)$$

where the second line relies on the fact that $|y + b| = y + b$ and $|y - b| = b - y$ for $|y| \leq b$. One can easily see, e.g., by differentiation, that $g_{i,\nu}$ is minimized at $y^* := -2\delta\nu\frac{1+\theta}{1-\theta}b$ which does satisfy $|y^*| \leq b$ given our assumption $\frac{1-\theta}{1+\theta} \geq 2\delta$. It follows that $\min_{y \in \mathcal{X}_i} g_{i,1}(y) = \min_{y \in \mathcal{X}_i} g_{i,-1}(y) = ab^2\left(\frac{1+3\theta}{4} - \delta^2\frac{(1+\theta)^2}{1-\theta}\right)$. Similarly, we have, for $y \in \mathcal{X}_i$,

$$g_{i,1}(y) + g_{i,-1}(y) = a \left( \frac{1-\theta}{2} y^2 + \frac{1+3\theta}{2} b^2 \right)$$

which is minimized at $y^* = 0$, where it takes value $ab^2 \frac{1+3\theta}{2}$. Putting it together,

$$\psi_i(\mathcal{G}_{\texttt{sc}}) = \min_{y \in \mathcal{X}_i}(g_{i,1}(y) + g_{i,-1}(y)) - \left(\min_{y \in \mathcal{X}_i} g_{i,1}(y) + \min_{y \in \mathcal{X}_i} g_{i,-1}(y)\right) = 2ab^2\delta^2 \frac{(1+\theta)^2}{1-\theta}.$$

Finally, $\psi(\mathcal{G}_{\texttt{sc}}) = \min_{i \in [d]} \psi_i(\mathcal{G}_{\texttt{sc}}) = 2ab^2\delta^2 \frac{(1+\theta)^2}{1-\theta} \geq \frac{2ab^2\delta^2}{1-\theta}$, as claimed. $\qquad\square$

### 4.5 Convex Lipschitz functions for $p \in [1, 2)$: Proof of Theorems 1, 4, and 7

We first prove Theorems 1 and 4, our lower bounds on optimization of convex functions for $p \in [1, 2)$ under privacy and communication constraints, respectively. We consider the class of functions $\mathcal{G}_{\texttt{c}}$ defined in (13) with parameters $a := 2B\delta/d^{1/q}$ and $b := D/(2d^{1/p})$. That is, $\mathcal{X} = \{x \in \mathbb{R}^d : \|x\|_\infty \leq D/(2d^{1/p})\}$ and

$$g_v(x) := \frac{2B\delta}{d^{1/q}} \sum_{i=1}^{d} \left| x(i) - \frac{v(i)D}{2d^{1/p}} \right| \qquad x \in \mathcal{X}, v \in \{-1, 1\}^d. \tag{17}$$

Note that the gradient of $g_v$ is equal to $-2B\delta v/d^{1/q}$ at every $x \in \mathcal{X}$.

For each $g_v$, consider the corresponding gradient oracle $O_v$ which outputs independent values for each coordinate, with the $i$th coordinate taking values $-B/d^{1/q}$ and $B/d^{1/q}$ with probabilities $(1 + 2\delta v(i))/2$ and $(1 - 2\delta v(i))/2$, respectively, for some parameter $\delta > 0$ to be suitably chosen later.

Clearly, $\mathcal{X} \in \mathbb{X}_p(D)$ and all the functions $g_v$ and the corresponding oracles $O_v$ belong to the convex function family $\mathcal{O}_{\texttt{c},p}$. We begin by noting that for $V$ distributed uniformly over $\{-1, 1\}^d$, we have

$$\sup_{\mathcal{X} \in \mathbb{X}_p(D)} \mathcal{E}^*(\mathcal{X}, \mathcal{O}_{\texttt{c},p}, T, \mathcal{W}_{\texttt{priv},\varepsilon}) \geq \mathbb{E}\left[g_V(x_T) - g_V(x_V^*)\right],$$

where the expectation is over $v$ as well as the randomness in $x_T$.

From Lemma 3 and 4, we have

$$\mathbb{E}\left[g_V(x_T) - g_V(x_V^*)\right] \geq \frac{d \cdot ab}{3} \cdot \left[1 - \sqrt{\frac{2}{d} \sum_{i=1}^{d} I(V(i) \wedge Y^T)}\right], \tag{18}$$

where $Y^T = (Y_1, ..., Y_T)$ are the channel outputs for the gradient estimates supplied by the oracle for the $T$ queries.

Next, we apply the average information bound from Section 4.3. To do so, observe that by the definition of our oracle, the oracle output at each time step is an independent draw from the product distribution $\mathbf{p}_v$ on $\Omega := \left\{-\frac{B}{d^{1/q}}, \frac{B}{d^{1/q}}\right\}^d$ (in particular, $\mathbf{p}_v$ is the same at each time step, as it does not depend on the query $x_t$ at time step $t$ to the oracle). We treat the output of the independent outputs of the oracle as i.i.d. samples $X_1, ..., X_T$ in Section 4.3 and the corresponding channel outputs as $Y^T$. We can check that, for every $i \in [d]$, we have

$$\frac{\mathbf{p}_{v \oplus i}(x)}{\mathbf{p}_v(x)} = \frac{1 + 2\delta v(i) \operatorname{sign}(x(i))}{1 - 2\delta v(i) \operatorname{sign}(x(i))} \tag{19}$$

for all $x \in \Omega$, and that Assumption 2 is satisfied with

$$\gamma := \frac{4\delta}{\sqrt{1 - 4\delta^2}}, \qquad \phi_{i,v}(x) := \frac{v(i) \operatorname{sign}(x(i)) + 2\delta}{\sqrt{1 - 4\delta^2}}. \tag{20}$$

Furthermore, noting that Assumption 3 always holds with

$$\kappa_{\mathcal{W}} = \max_{v \in \{-1,1\}^d} \max_{x \in \Omega} \max_{i \in [d]} \frac{\mathbf{p}_{v \oplus i}(x)}{\mathbf{p}_v(x)},$$

it is satisfied with $\kappa_{\mathcal{W}} = 2$ (regardless of $\mathcal{W}$), as long as $\delta \leq 1/6$, since the right-side above is bounded by 2 for such a $\delta$. Finally, Assumption 4, is also satisfied as $(\phi_{i,v}(X))_{i \in [d]}$ for $X \sim \mathbf{p}_v$ is $\sigma^2$-subgaussian for $\sigma^2 := \frac{1}{1-4\delta^2}$.

**Completing the proof of Theorem 1 (LDP constraints).** From Theorem 9 and the bounds derived above, we have

$$\sum_{i=1}^{d} I\big(V(i) \wedge Y^T\big) \leq T \cdot \frac{8\delta^2}{1 - 4\delta^2} \cdot e^\varepsilon (e^\varepsilon - 1)^2,$$

and therefore,

$$\sum_{i=1}^{d} I\big(V(i) \wedge Y^T\big) \leq c \cdot T\delta^2 \varepsilon^2,$$

where $c := 9e(e-1)^2$ (recalling that $\varepsilon \in (0, 1]$ and $\delta \leq 1/6$). Substituting this bound on the average mutual information in (18) along with the values of $a$ and $b$, we have

$$\mathbb{E}\left[g_V(x_T) - g_V(x_V^*)\right] \geq \frac{DB\delta}{3} \cdot \left[1 - \sqrt{\frac{2cT\delta^2 \varepsilon^2}{d}}\right].$$

Upon setting $\delta := \sqrt{\frac{d}{8cT\varepsilon^2}}$, we get

$$\mathbb{E}\left[g_V(x_T) - g_V(x_V^*)\right] \geq \frac{1}{12\sqrt{2c}} \cdot \frac{DB}{\sqrt{T}} \cdot \sqrt{\frac{d}{\varepsilon^2}},$$

where we require $T \geq \frac{9}{2c} \cdot \frac{d}{\varepsilon^2}$ in order to enforce $\delta \leq 1/6$. □

**Completing the proof of Theorem 4 (Communication constraints).** From Theorem 10 and $\gamma$, $\sigma$, and $\kappa_\mathcal{W}$ set as discussed above, we have

$$\sum_{i=1}^{d} I\big(V(i) \wedge Y^T\big) \leq \frac{32(\ln 2)}{(1 - 4\delta^2)^2} \cdot T\delta^2 r,$$

whereby, using $\delta \leq 1/6$,

$$\sum_{i=1}^{d} I\big(V(i) \wedge Y^T\big) \leq 29T\delta^2 r.$$

Substituting this bound on mutual information in (18) along with the values of $a$ and $b$, we have

$$\mathbb{E}\left[g_V(x_T) - g_V(x_V^*)\right] \geq \frac{DB\delta}{3} \cdot \left[1 - \frac{1}{\sqrt{d}} \cdot \sqrt{58T\delta^2 r}\right].$$

Setting $\delta := \sqrt{\frac{d}{232rT}}$, we finally get

$$\mathbb{E}\left[g_V(x_T) - g_V(x_V^*)\right] \geq \frac{1}{12\sqrt{58}} \cdot \frac{DB}{\sqrt{T}} \cdot \sqrt{\frac{d}{r}},$$

where we require $T \geq \frac{9}{58} \cdot \frac{d}{r}$ in order to enforce $\delta \leq 1/6$. □

**Completing the proof of Theorem 7 (Computational constraints).** Note that the sets $\mathcal{X}_i$s in Theorem 11 have $|\mathcal{X}_i| = 2$ for our oracle. Further,

$$\frac{\mathbf{p}_{v \oplus i}(X(i) = x(i))}{\mathbf{p}_v(X(i) = x(i))} = \frac{\mathbf{p}_{v \oplus i}(x)}{\mathbf{p}_v(x)} = \frac{1 + 2\delta v(i)\, \text{sign}(x(i))}{1 - 2\delta v(i)\, \text{sign}(x(i))} \leq 2,$$

when $\delta \leq 1/6$. Thus, the constant $C$ in Theorem 11 is less than 2, whereby

$$\sum_{i=1}^{d} I\big(V(i) \wedge Y^T\big) \leq \frac{16\delta^2}{1 - 4\delta^2} \cdot T,$$

whereby, using $\delta \leq 1/6$,

$$\sum_{i=1}^{d} I\big(V(i) \wedge Y^T\big) \leq 18T\delta^2.$$

Substituting this bound on mutual information in (18) along with the values of $a$ and $b$, we have
$$\mathbb{E}\left[g_V(x_T) - g_V(x_V^*)\right] \geq \frac{DB\delta}{3} \cdot \left[1 - \frac{1}{\sqrt{d}} \cdot \sqrt{36T\delta^2}\right].$$

Setting $\delta := \sqrt{\frac{d}{144T}}$, we finally get
$$\mathbb{E}\left[g_V(x_T) - g_V(x_V^*)\right] \geq \frac{1}{72} \cdot \frac{DB\sqrt{d}}{\sqrt{T}},$$

where we require $T \geq \frac{d}{4}$ in order to enforce $\delta \leq 1/6$.

## 4.6 Convex Lipschitz functions for $p \in [2,\infty]$: Proof of Theorems 2 and 5

Next, we establish Theorems 2 and 5, the analogous lower bounds on optimization of convex functions when $p \in [2,\infty)$. We again consider the class of functions $\mathcal{G}_{\mathsf{c}}$ defined in (13), this time with parameters $a := 2B\delta/d$ and $b := D/(2d^{1/p})$ That is, here $\mathcal{X} = \{x : \|x\|_\infty \leq D/(2d^{1/p})\}$ and
$$g_v(x) := \frac{2B\delta}{d} \sum_{i=1}^{d} \left| x(i) - \frac{v(i)D}{2d^{1/p}} \right|. \quad \forall x \in \mathcal{X}, v \in \{-1,1\}^d.$$

It follows that the gradient of $g_v$ is equal to $-2B\delta v/d$ at every $x \in \mathcal{X}$.

For each $g_v$, consider then the gradient oracle $O_v$ which outputs 0 in all but a randomly chosen coordinate; if that coordinate is $i$, it takes values $-B$ and $B$ with probabilities $\frac{1+2\delta v(i)}{2d}$ and $\frac{1-2\delta v(i)}{2d}$, respectively, for some parameter $\delta \in (0,1/6]$ to be suitably chosen later. Thus, the oracle is no longer a product distribution.

Clearly, $\mathcal{X} \in \mathbb{X}_p(D)$ and all the functions $g_v$ and the corresponding oracles $O_v$ belong to the convex function family $\mathcal{O}_{\mathsf{c},p}$. Proceeding as in Section 4.5, we get for a uniformly distributed $V$ that
$$\mathbb{E}\left[g_V(x_T) - g_V(x_V^*)\right] \geq \frac{DB\delta}{3d^{1/p}} \cdot \left[1 - \sqrt{\frac{1}{d} \sum_{i=1}^{d} 2I(V(i) \wedge Y^T)}\right]. \tag{21}$$

Further, proceeding as in the previous section to bound the average information, we note that the oracle outputs independent samples from the distribution $\mathbf{p}_v$ on $\Omega := \{-B, 0, B\}^d$ at each time. It can be checked easily that, for every $i \in [d]$, the expression of the ratio $\frac{\mathbf{P}_{v \oplus i}}{\mathbf{p}_v}$ given in (19) still holds (as only the denominators of the Bernoulli parameters have changed, and they cancel out in the ratio), and that Assumption 2 is satisfied with the following $\gamma$, $\phi_{i,v}$s:
$$\gamma := \frac{1}{\sqrt{d}} \cdot \frac{4\delta}{\sqrt{1-4\delta^2}}, \qquad \phi_{i,v}(x) := \sqrt{d} \cdot \frac{v(i)\,\mathrm{sign}(x(i)) + 2\delta}{\sqrt{1-4\delta^2}}. \tag{22}$$

Observe the difference with the expressions from the previous section (specifically, (20)), as the orthonormality assumption now crucially introduces a factor $1/\sqrt{d}$ in the value of $\gamma$. Finally, because we will enforce $\delta \leq 1/6$ we also can take $\kappa_{\mathcal{W}_{\mathsf{com},r}} = 2$ for the communication constraints, as before. We remark that $\phi_{i,v}(X)$ is no longer subgaussian.

**Completing the proof of Theorem 2 (LDP constraints).** From Theorem 9 and the value of $\gamma$ above, we get, analogously to the previous section,
$$\sum_{i=1}^{d} I\left(V(i) \wedge Y^T\right) \leq c \cdot \frac{T\delta^2\varepsilon^2}{d},$$

where $c := 9e(e-1)^2$ (recalling that $\varepsilon \in (0,1]$ and $\delta \leq 1/6$). Substituting this bound on mutual information in (21), we obtain
$$\mathbb{E}\left[g_V(x_T) - g_V(x_V^*)\right] \geq \frac{DB\delta}{3d^{1/p}} \left[1 - \sqrt{\frac{2cT\delta^2\varepsilon^2}{d^2}}\right].$$

Optimizing over $\delta$, we set $\delta := \sqrt{\frac{d^2}{8cT\varepsilon^2}}$ and get

$$\mathbb{E}\left[g_V(x_T) - g_V(x_V^*)\right] \geq \frac{1}{12\sqrt{2c}} \cdot \frac{DBd^{1/2-1/p}}{\sqrt{T}} \cdot \sqrt{\frac{d}{\varepsilon^2}},$$

where we require $T \geq \frac{9}{2c} \cdot \frac{d^2}{\varepsilon^2}$ in order to guarantee $\delta \leq 1/6$. This concludes the proof. $\square$

**Completing the proof of Theorem 5 (Communication constraints).** We prove the two parts of the lower bounds separately, starting with the first. From Theorem 10 and the setting of $\gamma$ and $\kappa_{\mathcal{W}}$ as above, we have

$$\sum_{i=1}^{d} I\left(V(i) \wedge Y^T\right) \leq \frac{16}{1-4\delta^2} \cdot T\delta^2 \frac{2^r \wedge d}{d},$$

whereby, using $\delta \leq 1/6$,

$$\sum_{i=1}^{d} I\left(V(i) \wedge Y^T\right) \leq 18T\delta^2 \frac{2^r \wedge d}{d}.$$

Substituting this bound on mutual information in (21), we have

$$\mathbb{E}\left[g_V(x_T) - g_V(x_V^*)\right] \geq \frac{DB\delta}{3d^{1/p}} \cdot \left[1 - \frac{1}{\sqrt{d}} \cdot \sqrt{36T\delta^2 \frac{2^r \wedge d}{d}}\right].$$

Setting $\delta := \sqrt{\frac{d^2}{144(2^r \wedge d)T}}$, we finally get

$$\mathbb{E}\left[g_V(x_T) - g_V(x_V^*)\right] \geq \frac{1}{72} \cdot \frac{DBd^{1/2-1/p}}{\sqrt{T}} \cdot \sqrt{\frac{d}{2^r \wedge d}},$$

where we require $T \geq \frac{1}{4} \cdot \frac{d^2}{2^r \wedge d}$ in order to guarantee $\delta \leq 1/6$.

The second bound follows by noting that the lower bound in Theorem 4 is still valid. Finally, since $\frac{d^2}{2^r \wedge d} \geq \frac{d}{r}$ for all $1 \leq r \leq d$, both bounds apply whenever $T = \Omega\left(\frac{d^2}{2^r \wedge d}\right)$, as claimed. $\square$

### 4.7 Strongly convex functions: Proof of Theorem 3, 6, and 8

Next, we establish our lower bounds on strongly convex optimization. We consider the class of functions $\mathcal{G}_{\mathrm{sc}}$ defined in (14) with parameters $a := B/(\sqrt{d}b)$ and $b := D/(2\sqrt{d})$. That is, $\mathcal{X} = \{x : \|x\|_\infty \leq D/(2\sqrt{d})\}$, and, for every $x \in \mathcal{X}$ and $v \in \{-1,1\}^d$,

$$g_v(x) := \frac{B}{b \cdot \sqrt{d}} \sum_{i=1}^{d} \frac{1+2\delta v(i)}{2} f_i^+(x) + \frac{1-2\delta v(i)}{2} f_i^-(x),$$

and

$$f_i^+(x) = \theta b|x(i) + b| + \frac{1-\theta}{4}(x(i)+b)^2 \text{ and } f_i^-(x) = \theta b|x(i) - b| + \frac{1-\theta}{4}(x(i)-b)^2.$$

Moreover, in order to ensure that the every $g_v$ is $\alpha$-strongly convex, we choose $\theta := 1 - \frac{4\alpha}{a}$ (so that $a\frac{1-\theta}{4} = \alpha$). It remains to specify $\delta$, which we will choose such that $0 < \delta \leq \frac{1}{2} \cdot \frac{1-\theta}{1+\theta}$ in the course of the proof.

For each $g_v$, consider the gradient oracle $O_v$ which on query $x$ outputs independent values for each coordinate, with the $i$th coordinate taking values $\frac{B}{b\sqrt{d}} \cdot \frac{\partial f_i^+(x)}{\partial x_i}$ and $\frac{B}{b\sqrt{d}} \cdot \frac{\partial f_i^-(x)}{\partial x_i}$ with probabilities $\frac{1+2\delta v(i))}{2}$ and $\frac{1-2\delta v(i)}{2}$, respectively.

Note that we have $\left|\frac{\partial f_i^+(x)}{\partial x_i}\right|, \left|\frac{\partial f_i^-(x)}{\partial x_i}\right| \leq b$ for all $x$ and $i$, and therefore the gradient estimate $\hat{g}(x)$ supplied by the oracle $O_v$ at $x$ satisfies $\|\hat{g}(x)\|_2^2 \leq B^2$ with probability one, for

every query $x \in \mathcal{X}$. Further, it is clear that $\mathcal{X} \in \mathbb{X}_2(D)$ and all the functions $g_v$ and the corresponding oracles $O_v$ belong to the strongly convex function family $\mathcal{O}_{\mathsf{sc}}$.

Using our assumption that $\delta \leq \frac{1}{2} \cdot \frac{1-\theta}{1+\theta}$, we obtain by Lemma 5

$$\psi(\mathcal{G}_{\mathsf{sc}}) \geq \frac{2ab^2\delta^2}{1-\theta} = \frac{2a^2b^2\delta^2}{4\alpha} = \frac{B^2\delta^2}{2d\alpha}, \tag{23}$$

where we first plug in $a(1-\theta) = 4\alpha$ and then substitute for $a$ and $b$.

**Completing the proof of Theorem 6 (Communication constraints).** By proceeding as in Section 4.5, from Lemma 3 and using the inequality (23) above, we have

$$\sup_{\mathcal{X} \in \mathbb{X}_2(D)} \mathcal{E}^*(\mathcal{X}, \mathcal{O}_{\mathsf{sc}}, T, \mathcal{W}_{\mathsf{com},r}) \geq \frac{B^2\delta^2}{12\alpha} \left[ 1 - \sqrt{\frac{2}{d} \sum_{i=1}^{d} I(V(i) \wedge Y^T)} \right]. \tag{24}$$

It remains to bound $\sum_{i=1}^{d} I(V(i) \wedge Y^T)$ to complete the proof. Note that unlike the proof in Section 4.5, the gradient estimates have different distributions for different $x$. However, for a point $x$ we can still express the gradient estimate $\hat{z}(x)$ of $g_v(x)$ given by $O_v$ as follows: abbreviating $f_i'^+(x) := \frac{\partial f_i^+(x)}{\partial x_i}$ and $f_i'^-(x) := \frac{\partial f_i^-(x)}{\partial x_i}$, we have

$$\hat{z}(x)(i) = aZ_i f_i'^+(x) + a(1-Z_i)f_i'^-(x), \tag{25}$$

where $Z_i \sim \mathrm{Ber}(1/2 + \delta v(i))$ and the $Z_i$'s are mutually independent. Thus, for a fixed $x$, $\hat{z}(x)$ can be viewed as a function of $\{Z_i\}_{i \in [d]}$. Furthermore, for a channel $W \in \mathcal{W}_{\mathsf{com},r}$ consider the channel $W'_x$ which first passes the Bernoulli vector $\{Z_i\}_{i \in [d]}$ through the function $\hat{z}(x)(i)$ and the resulting output is passed through the channel $W$. This composed channel $W_x$ belongs to $\mathcal{W}_{\mathsf{com},r}$, too.

Therefore, we can treat the independent copies of $Z \sim \mathbf{p}_v$ revealed by the oracle as i.i.d. random variables $X_1, ..., X_n$ in Section 4.3. Further, note that at time $t$, the query is for a point $x_t$ which is a random function of $Y^{t-1}$, and so, $Y^T$ can be viewed as the channel outputs with adaptively selected channels from $\mathcal{W}_{\mathsf{com},r}$. Thus, we can apply the bounds in Theorem 10.

Doing so, analogously to the computations in Section 4.5,[10] we get

$$\sum_{i=1}^{d} I(V(i) \wedge Y^T) \leq c\delta^2 rT,$$

for an appropriate constant $c$, which in view of (24) leads to

$$\sup_{\mathcal{X} \in \mathbb{X}_2(D)} \mathcal{E}^*(\mathcal{X}, \mathcal{O}_{\mathsf{sc}}, T, \mathcal{W}_{\mathsf{com},r}) \geq \frac{B^2\delta^2}{12\alpha} \cdot \left[ 1 - \frac{1}{\sqrt{d}} \cdot \sqrt{2cT\delta^2 r} \right] = \frac{1}{192c} \cdot \frac{B^2}{\alpha T} \cdot \frac{d}{r}$$

the last equality by setting $\delta := \sqrt{\frac{d}{8cTr}}$. Finally, observe that this choice of $\delta$ indeed satisfies $\delta < \frac{1}{2} \cdot \frac{1-\theta}{1+\theta}$, as long as $T \geq 2c \cdot \frac{B^2}{D^2} \cdot \frac{d}{\alpha^2 r}$. This completes the proof. $\qquad\square$

**Completing the proof of Theorem 3 (Privacy constraints).** Proceeding as in the proof of Theorem 6 above, we have the analogue of (24),

$$\sup_{\mathcal{X} \in \mathbb{X}_2(D)} \mathcal{E}^*(\mathcal{X}, \mathcal{O}_{\mathsf{sc}}, T, \mathcal{W}_{\mathsf{priv},\varepsilon}) \geq \frac{B^2\delta^2}{12\alpha} \left[ 1 - \sqrt{\frac{2}{d} \sum_{i=1}^{d} I(V(i) \wedge Y^T)} \right].$$

As stated in the proof of Theorem 6, the privatization of the gradient $\hat{z}(x)$ can be viewed as first preprocessing $\{Z_i\}_{i \in [d]}$ and the passing the preprocessed output through the LDP

---

[10]As we have, in both cases, unknown Bernoulli product distribution over $\{-1,1\}^d$ with bias vector $\frac{1}{2} + \delta v$.

channel. Such a composed channel also belongs to $\mathcal{W}_{\mathtt{priv},p}$. Thus, we can apply the bound in Theorem 9 and proceed as in the proof of Theorem 1 to obtain

$$\sum_{i=1}^{d} I\big(V(i) \wedge Y^T\big) \leq cT\delta^2 \varepsilon^2$$

where $c > 0$ is an absolute constant. Choosing $\delta := \sqrt{\frac{d}{8cT\varepsilon^2}}$, which makes $2\delta$ less than $\frac{1-\theta}{1+\theta}$ for $T \geq 2c \cdot \frac{B^2}{D^2} \cdot \frac{d}{\alpha^2 \varepsilon^2}$, for some universal positive constant $c$, then yields

$$\sup_{\mathcal{X} \in \mathbb{X}_2(D)} \mathcal{E}^*(\mathcal{X}, \mathcal{O}_{\mathtt{sc}}, T, \mathcal{W}_{\mathtt{priv},\varepsilon}) \geq c_0 \cdot \frac{B^2}{\alpha T} \cdot \frac{d}{\varepsilon^2}$$

for some absolute constant $c_0 > 0$, concluding the proof. $\qquad\square$

**Completing the proof of Theorem 8 (Computational constraints).** As before, we can get

$$\sup_{\mathcal{X} \in \mathbb{X}_2(D)} \mathcal{E}^*(\mathcal{X}, \mathcal{O}_{\mathtt{sc}}, T, \mathcal{W}_{\mathtt{obl}}) \geq \frac{B^2 \delta^2}{12\alpha} \left[1 - \sqrt{\frac{2}{d} \sum_{i=1}^{d} I(V(i) \wedge Y^T)}\right].$$

Recall that we can express the subgradient estimate as in (25). Note that for an oblivious sampling channel $W_t$ used at time $t$, specified by a probability vector $(p_j)_{j \in [d]}$, the output is given by

$$Y_i = (aZ_{J_t} f_{J_t}'^{+}(x) + a(1 - Z_{J_t}) f_{J_t}'^{-}(x)) e_{J_t},$$

where $J_t = j$ with probability $p_j$. To proceed, we observe that the Markov relation $V$—$\{Z_{J_t}, J_t\}_{t \in [T]}$—$Y^T$ holds. Indeed, we can confirm this by noting that $\{Z_{J_t}\}_{t \in [T]}$ are generated i.i.d. from $\mathbf{p}_V$ and, for each $t \in [T]$, $Y_t$ is a function of $(Y^{t-1}, Z_{J_t}, J_t)$ and a local randomness $U$ available only to the optimization algorithm which is independent jointly of $V$ and $\{Z_{J_t}, J_t\}_{t \in [T]}$. It follows that $Y^T$ itself is a function of $U$ and $\{Z_{J_t}, J_t\}_{t \in [T]}$, which gives

$$I\big(V \wedge Y^T \mid \{Z_{J_t}, J_t\}_{t \in [T]}\big) \leq I\big(V \wedge U \mid \{Z_{J_t}, J_t\}_{t \in [T]}\big) = 0. \qquad (26)$$

From the previous observation, we also get that the Markov relation $V(i)$—$\{Z_{J_t}, J_t\}_{t \in [T]}$—$Y^T$ holds for every $i \in [d]$. Thus, by the data processing inequality for mutual information, we have

$$\sum_{i=1}^{d} I\big(V(i) \wedge Y^T\big) \leq \sum_{i=1}^{d} I\big(V(i) \wedge \{Z_{J_t}, J_t\}_{t \in [T]}\big).$$

Now since vector $(Z_j)_{j \in [d]}$ is a Bernoulli vector, the mutual information on the right-side can be bounded by the same computation as in the proof of Theorem 7 using Theorem 11. This follows by observing that for all $t \in [T]$, $(Z_{J_t}, J_t)$ is a function of $Z_{J_t} e_{J_t}$, which in turn can be seen as a output of the oblivious sampling channel for an input vector $(Z_j)_{j \in [d]}$. Therefore, we have

$$\sum_{i=1}^{d} I\big(V(i) \wedge Y^T\big) \leq cT\delta^2$$

for an appropriate constant $c$ and $\delta \leq \frac{1}{6}$, which in view of (24) leads to

$$\sup_{\mathcal{X} \in \mathbb{X}_2(D)} \mathcal{E}^*(\mathcal{X}, \mathcal{O}_{\mathtt{sc}}, T, \mathcal{W}_{\mathtt{com},r}) \geq \frac{B^2 \delta^2}{12\alpha} \left[1 - \frac{1}{\sqrt{d}} \cdot \sqrt{2cT\delta^2}\right] = \frac{1}{c_0} \cdot \frac{dB^2}{\alpha T},$$

where the last identity is obtained by setting $\delta := c_1 \sqrt{\frac{d}{T}}$, where $c_0$ and $c_1$ are universal positive constants. Finally, observe that this choice of $\delta$ indeed satisfies $\delta < \frac{1}{2} \cdot \frac{1-\theta}{1+\theta}$, as long as $T \geq c_2 \cdot \frac{B^2}{D^2} \cdot \frac{d^2}{\alpha}$, for some universal positive constant $c_2$. This completes the proof. $\quad\square$

# 5 Adaptivity helps

In the previous sections we showed for information-constrained first-order optimization over the standard function and oracle classes, adaptive channel selection strategies offer no better minmax convergence guarantees than nonadaptive channel selection strategies. In all the cases, we made this claim in the minmax sense. Namely, we showed that for the worst-case function-oracle pair, adaptive schemes need not help. However, it does not imply that adaptivity does not help for *any* function-oracle pair. In fact, we now exhibit an interesting convex function class and associated oracle for which adaptivity can help.

Our example considers the oblivious sampling family $\mathcal{W}_{\texttt{obl}}$. Recall that in Randomized Coordinate Descent (RCD), the oracle returns the gradient along a single, randomly chosen coordinate [24, 28]. One can consider an adaptive version of this algorithm which allows to choose which coordinate to query the gradient for: we refer to this variant as *Adaptive Coordinate Descent* (ACD). We provide an example of a function class for which ACD has a strictly better performance than RCD, thereby showing that adaptive channel selection can help.

## 5.1 Mean estimation as an optimization problem.

The problem we consider entails a structured $\ell_2$ minimization. We first define $s$-block sparsity, which is needed to define our function class.

**Definition 4.** *A vector $v \in \mathbb{R}^d$ is $s$-block sparse if (i) there exists an $i$ such that $v_j = 0$ for all $j \notin \{is+1, \ldots, \min\{i(s+1), d\}\}$ and (ii) the nonzero coordinates have the same absolute value in $[0, 1]$. Let $\mathcal{B}_s$ be the set of all $s$-block sparse vectors in $d$ dimensions.*[11]

For $v \in \mathcal{B}_s$ and $\mathcal{X} = [-1, 1]^d$ let $f_v \colon \mathcal{X} \to \mathbb{R}$ be the function $f_v(x) = \|x - v\|_2^2$, $x \in \mathcal{X}$. Further, we associate with each function $f_v$ an oracle $O_v$ as follows. Let $X$ be a random variable over $\{-1, 1\}^d$ with $\mathbb{E}[X] = v$ (i.e., its mean is the $s$-block sparse vector $v$ parameterizing $f_v$). Moreover, we assume that each coordinate of $X$ is independent. The gradient estimate output of the oracle $O_v$ at $x$ and at time $t$ is $2(x - X_t)$, where $\{X_t\}_{t=1}^\infty$ are i.i.d. random variables with the same distribution as $X$. Note that the expected value of this gradient estimate is $\nabla f(x)$. Let $\mathcal{O}_{\texttt{blsp},s}$ denote the collection of pairs of functions and oracles described above.

As remarked earlier, we have fixed the class of oracles for our example. In our general formulation in Section 2.1, we did not even require the oracle to return independent outputs for different queries. The specific oracle above returns independent outputs for every query, and identically distributed outputs for the same query. Furthermore, the outputs are independent across the coordinates and each coordinate takes values $-1$ or $+1$. Interestingly, similar oracles were used in our lower bounds earlier.

Observe that the first-order optimization described above is the standard $\ell_2$ mean estimation problem cast as an optimization problem, since the function $f_v$ is minimized at $x^* := \mathbb{E}[X] = v$. Moreover, the essential information supplied by the oracle are the i.i.d. samples $X_t$ (since the algorithm already knows the queries $x$).

We will consider the block-sparse function and oracle class $\mathcal{O}_{\texttt{blsp},s}$ using the oblivious sampling channel family $\mathcal{W}_{\texttt{obl}}$ and show that adaptive channel selection strategies strictly outperform the nonadaptive ones. Towards that, we first derive a lower bound for nonadaptive strategies, and then present an adaptive scheme which improves over this bound.

Recall that $\mathcal{E}^{\text{NA}*}(\mathcal{X}, \mathcal{O}_{\texttt{blsp},s}, T, \mathcal{W}_{\texttt{obl}}) \geq \mathcal{E}^*(\mathcal{X}, \mathcal{O}_{\texttt{blsp},s}, T, \mathcal{W}_{\texttt{obl}})$. We will show a *strict* separation between the two quantities: for $s := \sqrt{d}$, the error incurred by any nonadaptive strategy is at least $\Omega(d^{3/2}/T)$, while there exists an adaptive strategy achieving error $O((d \log d)/T)$.

## 5.2 Lower bound for nonadaptive channel selection strategies

We show an $\Omega(ds/T)$ lower bound on the error for nonadaptive strategies.

---

[11] For simplicity, we assume throughout that $d/s$ is an integer.

**Theorem 12.** *Let $\mathcal{X} = [-1, 1]^d$. Then, there exists absolute constants $c_0, c_1, c_2 > 0$, such that for any $s \geq c_0$ and $T \geq c_1 d$, we have*

$$\mathcal{E}^{\mathrm{NA}*}(\mathcal{X}, \mathcal{O}_{\mathtt{blsp},s}, T, \mathcal{W}_{\mathtt{obl}}) \geq c_2 \cdot \frac{sd}{T}.$$

*Proof.* Let $\delta \in (0, 1/2]$ be a parameter to be determined in the course of the proof. Let $\mathcal{V}_s \subset \{-1, 1\}^s$ be a maximal $(s/4)$-packing in Hamming distance, i.e., a collection of vectors such that $d_H(v, v') > s/4$ for any two distinct $v, v' \in \mathcal{Z}$. By the Gilbert–Varshamov bound, we have $|\mathcal{V}_s| \geq 2^{cs}$ for some constant $c \in (0, 1)$. Now define the set $\mathcal{V} \subset \{-1, 1\}^d$ of $d$-dimensional $s$-block sparse vectors as follows:

$$\mathcal{V} = \bigcup_{i \in \{1, \ldots d/s\}} \{v^T = (v_1^T, \ldots, v_{d/s}^T) : v_i \in \mathcal{V}_s, v_j = 0 \in \mathbb{R}^s \; \forall j \neq i\}.$$

That is, $\mathcal{V}$ is the set of all $s$-block sparse vectors such that the non-sparse block contains all possible vectors from $\mathcal{V}_s$. From the definition, we immediately have $|\mathcal{V}| \geq \frac{d}{s} 2^{cs}$.

We will restrict ourselves to the subclass of functions $\mathcal{G}_{s,\delta} \subseteq \mathcal{G}_{\mathtt{blsp},s}$ consisting of all the functions of the form

$$f_{2\delta v}(x) = \|x - 2\delta v\|_2^2, \quad v \in \mathcal{V}.$$

Fix $v \in \mathcal{V}$. Clearly, the minimizer $x^*$ of $f_{2\delta v}$ is $2\delta v$, for which $f_{2\delta v}(x^*) = 0$, and therefore

$$f_{2\delta v}(x) - f_{2\delta v}(x^*) = \|x - 2\delta v\|_2^2.$$

Also, recall from the previous section that the oracle $O_{2\delta v}$ associated with $f_{2\delta v}$ will, upon query $x \in \mathbb{R}^d$, output the gradient estimate $2(x - X_v)$, where $X_v \in \{-1, 1\}^d$ is a random variable with mean $2\delta v$, whose distribution we get to specify. We will choose it as a product distribution over $\{-1, 1\}^d$, such that, for every $i \in [d]$,

$$\Pr(X_v(i) = 1) = \frac{1 + 2\delta v(i)}{2}, \quad \Pr(X_v(i) = -1) = \frac{1 - 2\delta v(i)}{2}. \tag{27}$$

We can verify that that $\mathbb{E}[X_v] = 2\delta v$.

We will use Fano's method to prove the lower bound. Fix any optimization algorithm $\pi$, and denote by $Y^T$ and $\hat{x} \in \mathbb{R}^d$ the corresponding transcript over the $T$ time steps and its eventual output, respectively. Let $V$ be distributed uniformly over $\mathcal{V}$. First, we relate the optimization error to the mutual information between $V$ and the messages $Y^T$:

**Claim 1.** *For $V$ and $Y^T$ as above, we have*

$$\mathbb{E}[f_{2\delta V}(x) - f_{2\delta V}(x^*)] \geq \frac{s\delta^2}{4}\left(1 - \frac{I(V \wedge Y^T) + 1}{cs + \log(d/s).}\right). \tag{28}$$

*Proof.* By Markov's inequality, we have

$$\mathbb{E}[f_{2\delta V}(x) - f_{2\delta V}(x^*)] = \mathbb{E}\left[\|x - 2\delta V\|_2^2\right] \geq \frac{s\delta^2}{4}\Pr\left(\|x - 2V\delta\|_2^2 \geq \frac{\sqrt{s}\delta}{2}\right),$$

where the expectation is over the uniform choice of $V$ and the randomness in choosing $x$.

Consider the multiple hypothesis testing problem of determining $V$ by observing $Y^T$. For this problem consider the estimator which, after running $\pi$ to obtain an approximate minimizer $\hat{x}$ of $f_{2\delta V}$, outputs the $V$ which is closest to the estimated $\hat{x}$, denoted by $V(\hat{x})$:

$$V(\hat{x}) := \operatorname*{argmin}_{u \in \mathcal{V}} \|\hat{x} - 2\delta u\|_2.$$

We will prove the following bound for the probability of error for this algorithm:

$$\Pr(V(\hat{x}) \neq V) \leq \Pr\left(\|\hat{x} - 2V\delta\|_2^2 \geq \sqrt{s}\delta/2\right).$$

To see this, recall that every distinct $u, u' \in \mathcal{V}$ satisfy $d_H(u, u') > s/4$, which implies $\|2\delta u - 2\delta u'\|_2 > \sqrt{s}\delta$. Therefore, whenever $\|\hat{x} - 2V\delta\|_2 < \sqrt{s}\delta/2$, the triangle inequality guarantees that, for every $u \in \mathcal{V}$ such that $u \neq V$,

$$\|\hat{x} - 2\delta u\|_2 \geq \|2\delta u - 2\delta V\|_2 - \|\hat{x} - 2\delta V\|_2 > \sqrt{s}\delta/2 > \|\hat{x} - 2\delta V\|_2.$$

It follows that

$$\Pr(V(\hat{x}) \neq V) \leq \Pr\left(\|\hat{x} - 2V\delta\|_2^2 \geq \sqrt{s}\delta/2\right),$$

as claimed. By Fano's inequality, we also have a lower bound on this error:

$$\Pr(V(\hat{x}) \neq V) \geq 1 - \frac{I(V \wedge Y^T) + 1}{\log |\mathcal{V}|}.$$

Putting the two together yields (28). $\qquad\square$

It remains to bound $I(V \wedge Y^T)$, which we do next.

**Claim 2.** *For $V$ and $Y^T$ as above, we have $I(V \wedge Y^T) \leq \frac{4\delta^2 sT}{d}$.*

*Proof.* Since $\pi$ is a nonadaptive protocol, the random variables $Y^T$ are independent (albeit not necessarily identically distributed). Therefore, by similar arguments as in proving (26) and denoting as before by $e_1, \ldots, e_d$ the standard basis vectors, we have

$$I(V \wedge Y^T) \leq \sum_{t=1}^{T} I(V \wedge X_V(J_t)e_{J_t}),$$

where $J_t = i$ with probability $p_i$, for all $i \in [d]$, and $J_{t_1}$ is independent of $J_{t_2}$.

We will derive a uniform bound for $I(V \wedge X_V(J_t)e_{J_t})$ for all $t \in [T]$. To do so, fix any $t \in [T]$, and denote by $W \in \mathcal{W}_{\text{obl}}$ the channel used as the $t$th time step and by $(p_i)_{i \in [d]}$ its corresponding distribution over coordinates. Denoting by $P_{X_{v'}}$ the product distribution described in (27) (when the underlying vector is $v'$) and recalling the definition of a channel in $\mathcal{W}_{\text{obl}}$, we can rewrite $P_{X_v(J_t)e_{J_t}|v'}$, the conditional pmf of $X_v(J_t)e_{J_t}$ given $V = v'$, as follows:

$$P_{X_v(J_t)e_{J_t}|v'}(e_i)) = p_i \cdot P_{X_{v'}(i)}(1) = p_i \cdot \frac{1 + 2\delta v'(i)}{2},$$

$$P_{X_V(J_t)e_{J_t}|v'}(-e_i) = p_i \cdot P_{X_{v'}(i)}(-1) = p_i \cdot \frac{1 - 2\delta v'(i)}{2},$$

for all $i \in [d]$. (In particular, $P_{X_v(J_t)e_{J_t}}$ is supported on $2d$ elements.)

Then, by joint-convexity of $D(P\|Q)$, we have

$$
\begin{aligned}
I(V \wedge X_v(J_t)e_{J_t}) &= \sum_{v' \in \mathcal{V}} P_V(v') D(P_{X_v(J_t)e_{J_t}|v'} \| P_{X_v(J_t)e_{J_t}}) \\
&\leq \sum_{v' \in \mathcal{V}} P_V(v') \sum_{i \in [d]} p_i D(P_{X_{v'}(i)} \| \sum_{v' \in \mathcal{V}} P_V(v') P_{X_{v'}(i)}) \\
&= \sum_{i \in [d]} p_i \sum_{v' \in \mathcal{V}} P_V(v') D(P_{X_{v'}(i)} \| \sum_{v' \in \mathcal{V}} P_V(v') P_{X_{v'}(i)}).
\end{aligned}
$$

Fixing $i \in [d]$, we now use the fact that

$$\sum_{v' \in \mathcal{V}} P_V(v') D(P_{X_{v'}(i)} \| \sum_{v' \in \mathcal{V}} P_V(v') P_{X_{v'}(i)}) \leq \sum_{v' \in \mathcal{V}} P_V(v') D(P_{X_{v'}(i)} \| Q),$$

for every $Q$ with support $\{-1, 1\}$. Choosing $Q$ as the uniform distribution over $\{-1, 1\}$, it then suffices to bound $\sum_{v' \in \mathcal{V}} P_V(v') D(P_{X_{v'}(i)} \| Q)$.

Note that $D(P_{X_{v'}(i)} \| Q) = 0$ unless $i$ belongs to the block of $s$ non-zero coordinates of $v'$. When $i$ belongs to that block, however, we get by upper bounding KL divergence by chi-square divergence that

$$D(P_{X_{v'}(i)} \| Q) \leq \sum_{x \in \{-1, 1\}} \frac{\left(P_{X_{v'}(i)}(x) - Q(x)\right)^2}{Q(x)} = \sum_{x \in \{-1, 1\}} \frac{\left(\frac{1 + 2\delta v'(i)x}{2} - \frac{1}{2}\right)^2}{1/2} = 4\delta^2.$$

Since $V$ is drawn uniformly at random, the probability (over $V$) that the block to which $i$ belongs is the non-sparse one is $s/d$. Consequently, $\sum_{v' \in \mathcal{V}} P_V(v') D(P_{X_{v'}(i)} \| Q) \leq \frac{4\delta^2 s}{d}$. As this holds for every $i \in [d]$, plugging this in our bound for $I(V \wedge Y_t)$ leads to

$$I(V \wedge Y_t) \leq \sum_{i \in [d]} p_i \sum_{v' \in \mathcal{V}} P_V(v') D(P_{X_{v'}(i)} \| Q) \leq \sum_{i \in [d]} p_i \cdot \frac{4\delta^2 s}{d} = \frac{4\delta^2 s}{d}.$$

Summing over all $t \in [T]$ then proves the claim. $\qquad\square$

In order to conclude the proof, we combine Claims 1 and 2, to obtain

$$\mathbb{E}\left[f_V(x) - f_V(x^*)\right] \geq s\delta^2 \left(1 - \frac{4T\delta^2 s/d + 1}{cs + \log(d/s)}\right)$$
$$\geq s\delta^2 \left(1 - \frac{4T\delta^2 s/d + 1}{cs}\right)$$
$$\geq s\delta^2 \left(1 - \frac{8T\delta^2}{cd}\right)$$

where the final inequality holds for $\delta^2 \geq \frac{d}{4Ts}$. We now choose $\delta^2 = \frac{cd}{16T}$, which is a valid choice for $s \geq \frac{4}{c}$, to get

$$\mathbb{E}\left[f_V(x) - f_V(x^*)\right] \geq \frac{c}{32} \cdot \frac{sd}{T},$$

where we require $T \geq \frac{cd}{4}$ to ensure $\delta^2 \leq \frac{1}{4}$, which, in turn, is essential for (27) to define a valid pmf. $\qquad\square$

## 5.3 Adaptivity helps

We now prove a $O((d \log(d/s) + s^2)/T)$ upper bound on the error for adaptive strategies, by exhibiting a specific adaptive channel selection strategy and optimization procedure we term *Adaptive Coordinate Descent* (ACD), denoted $\pi_{\texttt{ACD}}$.

First, note that the only new information that the oracles present at each iteration is about the random variable $X$ with $\mathbb{E}[X] = v$ underlying the oracle associated with some function $f_v(x) = \|x - v\|_2^2$ in our family $\mathcal{O}_{\texttt{blsp},s}$. Thus, the problem at hand becomes that of estimating the mean $v$ using independent copies of $X$. See Algorithm 1 for a detailed description.

Keeping this in mind, our adaptive channel selection strategy is divided in two phases, each making $T/2$ queries to the oracle:[12] the *exploration phase* and the *exploitation phase*. In the exploration phase, we select each block's first coordinate as a representative coordinate for that block and query each representative coordinate $Ts/(2d)$ times. At the end of this phase, an estimate of the mean is formed for each representative coordinate. Next, we select the block whose representative coordinate has the sample mean with the highest absolute value. Then, in the exploitation phase each coordinate of the selected block is queried $T/(2s)$ times.

Our optimization algorithm estimates the means of coordinates in the selected block using the sample mean of the values received in the exploitation phase. For the rest of the coordinates, the mean estimate is zero. Finally, our algorithm returns the overall estimated mean vector as the estimated minimizer of the function.

Recall that in RCD, the oracle returns the gradient along a randomly chosen coordinate. In contrast, ACD gets gradient for a particular coordinate in each round, and the choice of the coordinates used in the exploitation phase depends on the observations of the exploration phase. Also, we note that it is possible to interpret our procedure as a coordinate descent algorithm. However, for the ease of presentation, we simply retain the form above.

The performance of $\pi_{\texttt{ACD}}$ is characterized by the result below.

---

[12]We assume for simplicity that $T/2$, $Ts/(2d)$, and $T/(2s)$ are integers.

---
**Algorithm 1:** Adaptive Coordinate Descent $\pi_{\texttt{ACD}}$
---
/* Exploration phase:  the first $T/2$ oracle queries                */
**for** $i = 1$ **to** $d/s$ **do**

   $T_1 \leftarrow 1 + (i-1) \cdot \frac{Ts}{2d}$, $T_2 \leftarrow i \cdot \frac{Ts}{2d}$

   **for** $t = T_1$ **to** $T_2$ **do**

      Sample $X_t((i-1) \cdot s + 1)$, the $((i-1) \cdot s + 1)$th coordinate of the gradient
       estimate at time $t$

      Query the oracle for arbitrary $x \in [-1, 1]^d$

      Sample the $((i-1) \cdot s + 1)$th coordinate of the gradient estimate at time $t$

      Recover $X_t((i-1) \cdot s + 1)$ from the $((i-1) \cdot s + 1)$th coordinate of the gradient
       estimate

   **end**

   Compute

$$\hat{X}(i) \leftarrow \sum_{t=T_1}^{T_2} X_t((i-1) \cdot s + 1)$$

   ;

**end**

Set

$$i^* \leftarrow \arg\max_{i \in [d/s]} |\hat{X}(i)|$$

 and $\mathcal{I} \leftarrow \{i^*, \ldots, i^* + (s-1)\}$.

/* Exploitation phase:  the last $T/2$ oracle queries                */
**for** $i \in \mathcal{I}$ **do**

   Set $T_1 \leftarrow T/2 + 1 + (i-1) \cdot \frac{T}{2s}$ and $T_2 \leftarrow T/2 + i \cdot \frac{T}{2s}$

   **for** $t = T_1$ **to** $T_2$ **do**

      Query the oracle for arbitrary $x \in [-1, 1]^d$

      Sample the $i$th coordinate of the gradient estimate at time $t$

      Recover $X_t(i)$ from the $i$th coordinate of the gradient estimate

   **end**

   Compute

$$\hat{Y}(i) \leftarrow \frac{2s}{T} \sum_{t=T_1}^{T_2} X_t(i)$$

**end**

**for** $i \in [d] \setminus \mathcal{I}$ **do**

   $\hat{Y}(i) \leftarrow 0$

**end**

**Result:** $\hat{Y} = [\hat{Y}(1), \ldots, \hat{Y}(d)]^T$

---

**Theorem 13.** *Fix any $1 \leq s \leq d$, and $(f, O) \in \mathcal{O}_{\texttt{blsp},s}$.[13] Let $\hat{Y} \in \mathbb{R}^d$ be the point returned by Algorithm 1 after $T$ oracle queries to $O$. Then,*

$$\mathbb{E}\left[f(\hat{Y})\right] \leq \frac{36d \ln \frac{d}{s} + 2s^2}{T},$$

*Proof.* Fix $(f, O)$ as in the statement, so that $f$ is parameterized by some $s$-block sparse vector $v \in [-1, 1]^d$, with $f(x) = \|x - v\|_2^2$; and $O$ corresponds to the distribution of some random variable $X$ over $\{-1, 1\}^d$ with mean $\mathbb{E}[X] = v$. For simplicity, and without loss of generality, we assume that the block of non-sparse coordinates for the mean vector is $\{1, \ldots, s\}$. Further, let[14] $\delta := \mathbb{E}[X(1)]$. Using the same notation as in the description of

---

[13] That is, $f(x) = f_v(x) = \|x - v\|^2$ for some $v$ with block sparsity structure and $O$ gives independent copies of random variable $X$ with $\mathbb{E}[X] = v$.

[14] Recall from Definition 4 that all the non-zero mean coordinates have the same mean value in absolute value. Therefore, $|\mathbb{E}[X(i)]| = |\delta|$ for all $i \in \{1, \ldots, s\}$.

Algorithm 1, denote by $\mathcal{I}$ the index of the coordinates in the block selected by the algorithm. Moreover, let $\mathcal{J} := [d] \setminus \mathcal{I}$ be the set of remaining coordinates. We can rewrite the error as

$$\mathbb{E}\left[f(\hat{Y})\right] = \mathbb{E}\left[\|\hat{Y} - v\|_2^2\right] = \mathbb{E}\left[\sum_{i \in \mathcal{I}}(\hat{Y}(i) - v(i))^2\right] + \mathbb{E}\left[\sum_{i \in \mathcal{J}}(\hat{Y}(i) - v(i))^2\right]. \tag{29}$$

We will bound both terms separately. To handle the first, recall that, for all $i \in \mathcal{I}$, we have $\hat{Y}(i) = \frac{2s}{T}\sum_{t=T_1}^{T_2} X_t(i)$, where $T_1 = \frac{T}{2} + 1 + (i-1) \cdot \frac{T}{2s}$ and $T_2 = \frac{T}{2} + i \cdot \frac{T}{2s}$. Therefore, for all $i \in [d]$,

$$\mathbb{E}\left[(\hat{Y}(i) - v(i))^2 \mathbf{1}_{\mathcal{I}}(i) \mid \mathcal{I}\right] = \frac{2s}{T}\mathbb{E}\left[(X_{T_1}(i) - v(i))^2\right]\mathbf{1}_{\mathcal{I}}(i) \leq \frac{2s}{T}\mathbb{E}\left[X_{T_1}(i)^2\right]\mathbf{1}_{\mathcal{I}}(i) \leq \frac{2s}{T}\mathbf{1}_{\mathcal{I}}(i),$$

where the first equality follows from the fact that the sequence of random vectors $\{X_t\}_{t=T/2+1}^{T}$ is i.i.d. and independent of the random set $\mathcal{I}$, along with the fact that $\mathbb{E}[X_{T_1}(i)] = v(i)$; and the second inequality is because $X_{T_1}(i) \in [-1, 1]$. Since $|\mathcal{I}| = s$, by the law of total expectation we get

$$\mathbb{E}\left[\sum_{i \in \mathcal{I}}(\hat{Y}(i) - v(i))^2\right] = \sum_{i=1}^{d}\mathbb{E}\left[(\hat{Y}(i) - v(i))^2\mathbf{1}_{\mathcal{I}}(i)\right] \leq \frac{2s^2}{T}. \tag{30}$$

We claim that the second term of the RHS can be bounded as follows:

$$\mathbb{E}\left[\sum_{i \in \mathcal{J}}(\hat{Y}(i) - v(i))^2\right] \leq \frac{36d}{T}\ln\frac{d}{s} \tag{31}$$

To see why, set $R := \frac{36d}{s}\ln\frac{d}{s}$, so that our goal is to show that $\mathbb{E}\left[\sum_{i \in \mathcal{J}}(\hat{Y}(i) - v(i))^2\right] \leq sR/T$. First, for all $i \in \mathcal{J}$, $\hat{Y}(i) = 0$, and so we have

$$\mathbb{E}\left[\sum_{i \in \mathcal{J}}(\hat{Y}(i) - v(i))^2\right] = \mathbb{E}\left[\sum_{i \in \mathcal{J}}v(i)^2\right] = s\delta^2 \Pr(\mathcal{I} \neq \{1, \ldots, s\}).$$

The last equality follows from the fact that if $\mathcal{I}$ is the correct block (which we assumed was $\{1, \ldots, s\}$), then $\mathcal{J}$ only contains coordinates $i$ for which the mean $v(i) = v_i = 0$; while if $\mathcal{I}$ is not the correct block, then all $s$ coordinates of that block are in $\mathcal{J}$, and each of them has $|v(i)| = |\delta|$.

If $|\delta| \leq \sqrt{R/T}$, we are done, as then $s\delta^2 \Pr(\mathcal{I} \neq \{1, \ldots, s\}) \leq sR/T$, which is what we wanted. Thus, we hereafter assume $|\delta| > \sqrt{R/T}$ and want to bound $\Pr(\mathcal{I} \neq \{1, \ldots, s\})$, which by the description of our algorithm is exactly the probability that $i^* \neq 1$. That is,

$$\Pr\left(|\hat{X}(1)| \leq \max_{2 \leq i \leq d/s}|\hat{X}(i)|\right),$$

where $\hat{X}(1), \hat{X}(2), \ldots, \hat{X}(d/s)$ are independent random variables, with $\hat{X}(2), \ldots, \hat{X}(d/s)$ being identically distributed as the sum of $N := \frac{Ts}{2d}$ independent 1-subgaussian r.v.'s and $\hat{X}(1)$ being the sum of $N$ i.i.d. random variables in $[-1, 1]$ with mean $\delta$. On the one hand, by a standard argument (see for instance [8]), one can check that

$$\mathbb{E}\left[\max_{2 \leq i \leq d/s}|\hat{X}(i)|\right] \leq \sqrt{2N\ln\frac{d}{s}} < \frac{1}{3}N\sqrt{\frac{R}{T}}$$

where the last inequality used our setting of $R \geq \frac{36d}{s}\ln\frac{d}{s}$. On the other hand,

$$\left|\mathbb{E}\left[\hat{X}(1)\right]\right| = N|\delta| > N\sqrt{\frac{R}{T}}$$

and therefore we have

$$\Pr\left(|\hat{X}(1)| \leq \max_{2 \leq i \leq d/s} |\hat{X}(i)|\right)$$

$$\leq \Pr\left(\left\{|\hat{X}(1)| \leq \frac{2}{3}N|\delta|\right\} \cup \left\{\max_{2 \leq i \leq d/s} |\hat{X}(i)| \geq \frac{2}{3}N|\delta|\right\}\right)$$

$$\leq \Pr\left(|\hat{X}(1)| \leq \frac{2}{3}N|\delta|\right) + \Pr\left(\max_{2 \leq i \leq d/s} |\hat{X}(i)| \geq \frac{2}{3}N|\delta|\right)$$

$$\leq \Pr\left(|\hat{X}(1)| \leq \frac{2}{3}\left|\mathbb{E}\left[\hat{X}(1)\right]\right|\right) + \Pr\left(\max_{2 \leq i \leq d/s} |\hat{X}(i)| > \sqrt{2N \ln(d/s)} + \frac{1}{3}N|\delta|\right)$$

We handle both terms separately. By symmetry, we can assume without loss of generality that $\mathbb{E}\left[\hat{X}(1)\right] \geq 0$, and so

$$\Pr\left(|\hat{X}(1)| \leq \frac{2}{3}\left|\mathbb{E}\left[\hat{X}(1)\right]\right|\right) \leq \Pr\left(\hat{X}(1) \leq \frac{2}{3}\mathbb{E}\left[\hat{X}(1)\right]\right) \leq e^{-\frac{\mathbb{E}[\hat{X}(1)]^2}{18N}} = e^{-\delta^2 N/18}$$

by a Hoeffding bound. Note that we then have

$$s\delta^2 \Pr\left(|\hat{X}(1)| \leq \frac{2}{3}\left|\mathbb{E}\left[\hat{X}(1)\right]\right|\right) \leq s\delta^2 e^{-\delta^2 N/18} = \frac{sR}{T} \cdot \frac{36d}{sR} \cdot \frac{\delta^2 N}{18} e^{-\delta^2 N/18}$$

$$\leq \frac{sR}{T} \cdot e^{-1}$$

since $N = Ts/2d$, $R \geq 36d/s$.

Turning to the second term, by a standard concentration bound for the maximum of sub-gaussian r.v.'s and using the fact that each $\hat{X}(i)$, for $i \geq 2$, is $N$-subgaussian, we get

$$\Pr\left(\max_{2 \leq i \leq d/s} |\hat{X}(i)| > \sqrt{2N \ln(d/s)} + \frac{1}{3}N|\delta|\right) \leq e^{-\frac{(\delta N/3)^2}{2N}} = e^{-\delta^2 N/18}$$

and we conclude as before that

$$s\delta^2 \Pr\left(\max_{2 \leq i \leq d/s} |\hat{X}(i)| > \sqrt{2N \ln(d/s)} + \frac{1}{3}N|\delta|\right) \leq \frac{sR}{eT}.$$

This shows that, in this case,

$$s\delta^2 \Pr(\mathcal{I} \neq \{1, \dots, s\}) \leq 2e^{-1} \cdot sR/T \leq sR/T \tag{32}$$

as well. Plugging (30) and (32) in (29), we get $\mathbb{E}\left[\|\hat{Y} - \mu\|_2^2\right] \leq \frac{36d \ln(d/s) + 2s^2}{T}$, proving the theorem. $\qquad \square$

Combining Theorems 12 and 13, for $\mathcal{X} = [-1, 1]^d$ and $s = \sqrt{d}$ we obtain a strict separation between nonadaptive and adaptive strategies:

$$\mathcal{E}^{\text{NA}*}(\mathcal{X}, \mathcal{O}_{\texttt{blsp},s}, T, \mathcal{W}_{\texttt{obl}}) \gtrsim \frac{d^{3/2}}{T}, \qquad \text{but} \qquad \mathcal{E}^*(\mathcal{X}, \mathcal{O}_{\texttt{blsp},s}, T, \mathcal{W}_{\texttt{obl}}) \leq \frac{20d \ln d}{T}$$

for $T = \Omega(d)$. Note that the separation between adaptive and nonadaptive schemes hold for all $\log d \ll s \ll d$, but the multiplicative gain in convergence rate is maximized for $s \approx \sqrt{d}$.

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

# A    Proof of Lemma 1

From the strong convexity of $f$, we have

$$f\left(\frac{x+y}{2}\right) - \frac{\alpha}{2}\left\|\frac{x+y}{2}\right\|_2^2 \le \frac{1}{2}f(x) - \frac{\alpha}{4}\|x\|_2^2 + \frac{1}{2}f(y) - \frac{\alpha}{4}\|y\|_2^2, \quad \forall x, y \in \mathcal{X},$$

which upon reorganizing and using the fact that $2\|x\|_2^2 + 2\|x\|_2^2 - \|x+y\|_2^2 = \|x-y\|_2^2$ can be seen to be equivalent to

$$\frac{\alpha}{4}\|x-y\|_2^2 \le f(x) + f(y) - 2f\left(\frac{x+y}{2}\right).$$

Further, by Lipschitz continuity of $f$ in the $\ell_2$ norm, we have

$$f(x) + f(y) - 2f\left(\frac{x+y}{2}\right) \le B\|x-y\|_2.$$

Upon combining the previous two bounds, we obtain

$$\frac{\alpha}{4} \le \frac{f(x) + f(y) - 2f\left(\frac{x+y}{2}\right)}{\|x-y\|_2^2} \le \frac{B}{\|x-y\|_2},$$

which completes the proof upon substituting $y = 0$ and $x$ such that $|x(i)| = D$ for all $i \in [d]$ giving $\|x-y\|_2 = d^{1/2}D$.

## B  Upper Bounds for $p = 1$ under communication constraints.

We now provide a scheme which matches the lower bound of Theorem 4 for $\ell_1$ norm and $r$-bits communication constraints, for optimization for the family of convex functions. Our scheme divides the entire horizon of $T$ iterations into $Tr/d$ different phases. For any phase $t \in [Tr/d]$, the same point $x_t$ in the domain is queried $d/r$ times. For each of the $d/r$ queries in a phase, we use $r$-bit quantizers to quantize different coordinates of the subgradient output. At a high level, we want to use these $r$ bits to send 1 bit each for $r$ different coordinates, sending 1 bit for each coordinate across the phases. However, there is one technical difficulty. We have not assumed that making queries for the same point gives identically distributed random variables. We circumvent this difficulty using random permutations to create unbiased estimates for the subgradients.

Specifically, for a permutation $\sigma\colon [d] \to [d]$ chosen uniformly at random using public randomness, we select the coordinates $\sigma(1 + (i-1) \cdot r)$ to $\sigma(i \cdot r)$ of the subgradient estimate $\hat{g}_i$ supplied by the oracle for the $i$th query in the $t$th phase (i.e., $i$th time we query the point $x_t$) and quantize all of these coordinates using an 1-bit unbiased quantizer for the interval $[-B, B]$. Note that such a quantizer can be formed since $\|\hat{g}_i\|_\infty \le B$.

Using this procedure, the quantized gradient for every query in each phase can be stored in $r$ bits. Furthermore, using all the $d/r$ quantized estimates received in a phase, we can create an estimate of the subgradient by simply adding all the estimates. Denote by $\bar{Q}_t$ our subgradient estimate in the $t$th phase. Then,

$$\bar{Q}_t = \sum_{i=1}^{d} Q_{\pi(i)}(\hat{g}_i) e_{\pi(i)},$$

where $\hat{g}_i$ is the subgradient estimate returned by the oracle when we query $x_t$ for the $i$th time and $Q_i$ is a 1-bit unbiased estimator of the $i$th coordinate of gradient estimate given below: For all vectors $g$, such that $\|g\|_\infty \le B$, we have

$$Q_i(g) = \begin{cases} B & \text{w.p.} & \frac{g(i)+B}{2B} \\ -B & \text{w.p.} & \frac{B-g(i)}{2B} \end{cases}.$$

Then, we use $\bar{Q}_t$ to update $x_t$ to $x_{t+1}$ using stochastic mirror descent with mirror map

$$\phi_a(x) \colon = \frac{\|x\|_a^2}{a-1},$$

where $a = \frac{2\log d}{2\log d - 1}$. Recall that for a mirror map $\Phi$, the Bregman divergence associated with $\Phi$ is defined as

$$D_\Phi(x, y) \colon = \Phi(x) - \Phi(y) - \langle \nabla\Phi(y), x - y\rangle.$$

**Algorithm 2:** $\pi^*$ Optimal Scheme for Communication constrained optimization for $\ell_1$ convex family

---

**for** $t = 1$ **to** $Tr/d$ **do**
    **for** $i = 1$ **to** $d/r$ **do**

        At Center:
        Query the oracle for $x_t$

        At Oracle:
        Output the $r$-bit vector of 1-bit unbiased estimates of the $r$ coordinates
        $\{1 + (i-1) \cdot r, \ldots, i \cdot r\}$ of $\hat{g}_i(x_t)$ given by

$$\bar{Q}_t \leftarrow \sum_{j=1+(i-1)\cdot r}^{i\cdot r} Q_{\pi(j)}(\hat{g}_i(x_t)) e_{\pi(j)}$$

        At Center:
        $x_{t+1} \leftarrow \arg\min_{x \in \mathcal{X}}(\eta_t \langle x, \bar{Q}_t \rangle) + D_{\Phi_a}(x, x_t))$
    **end**

    **Result:** $\frac{\sum_{i=1}^{T} x_t}{T}$
**end**

---

**Theorem 14.** *For $r \in \mathbb{N}$, we have*

$$\sup_{\mathcal{X} \in \mathbb{X}_1(D)} \mathcal{E}^*(\mathcal{X}, \mathcal{O}_{\mathsf{c},1}, T, \mathcal{W}_{\mathsf{com},r}) \leq \frac{c_0 D B \sqrt{\log d}}{\sqrt{T}} \cdot \sqrt{\frac{d}{d \wedge r}}$$

*for every $D > 0$.*

*Proof.* Note that our first order optimization algorithm $\pi^*$ uses $Tr/d$ iterations. Moreover, the subgradient estimates $\bar{Q}_t$ are unbiased and have their infinity norm bounded by $B$. Namely, we have obtained an unbiased subgradient oracle which produces estimates with infinity norm bounded by $B$. Thus, using the standard analysis of mirror descent using noisy subgradient oracle for optimization over an $\ell_1$ ball with mirror map $\phi_a(x) := \frac{\|x\|_a^2}{a-1}$ (see Remark 1), the proof is complete. $\qquad\square$