# OpenReview forum: "Information-constrained optimization: can adaptive processing of gradients help?"
_NeurIPS.cc/2021/Conference — NeurIPS 2021 Poster_

### Official Review · Reviewer_Z9A8 · 2021-07-15

**Rating:** 6
**Confidence:** 2

**Summary:**

Here, all the references are to the supplementary material section.

This work studies the problem of first-order optimization under local constraints, where at each step an agent computes a sub-gradient and the sub-gradient should pass through a channel. Then, output of the channel will be available for the optimization algorithm. The act of the channel is to constraint the information of the sub-gradient. Their idea is to construct an adaptive method, where at each step the computed sub-gradient depends on the previous outputs of the channel, in advance. They call this approach adaptive and the main goal of the authors is to investigate whether there is any gain in using adaptive methods compared to nonadaptive ones.

Assumptions on the objective function: Denote the objective function as $f$, which is defined on a compact subset of $\mathbb{R}^d$. They studied the above problem in the case that:
1. $f$ is Lipschitz with respect to $\ell_p$-norm, where $p \geq 1$.
2. $f$ is (strongly) convex.

Different types of information constraints: Performance of an adaptive method has been studied in this paper over three different types of information constraints:
1. Local differential privacy (defined in page 7)
2. Communication constraints (defined in page 8)
3. Computational constraints (defined in page 8)

Contributions: They used a novel technique for obtaining lower bounds by using Asoud's lemma (outlined in the lines 80 to 87, on page 3 and Section 4) for the adaptive methods. By comparing their derived lower bounds with the upper bounds for the non-adaptive scheme, they concluded that there is no gain in using adaptive methods, in general setting.
At the end, they studied adaptivity for the special case of structured $\ell$_2 minimization (defined in Section 5.1, page 32.), for computational constraints (oblivious channels). First they derived a lower bound for non-adaptive setting, then they proposed an adaptive algorithm (Algorithm 1, page 28) and concluded that adaptive methods significantly outperform non-adaptive ones in this specific example.

**Ethical Concerns:**

The authors briefly discuss the limitations of their work. I also pointed out some, along with suggestions to improve the paper in the Major Review section.

**Limitations And Societal Impact:**

The authors briefly discuss the limitations of their work. I also pointed out some, along with suggestions to improve the paper in the Main Review section.

Since the paper is quite technical, I except no direct social impact at this point.

**Main Review:**

1. All the theoretical claims are well supported.
2. The paper is well written. The proofs are mathematically clear. Also, the authors provide an adequate literature review and give enough comparison between their results and the prior works.
3. I am not convinced of the motivation of the work. What is the motivation of the authors to study adaptive techniques in LDP and communication constraints? Are there important examples of functions for these constraints, where adaptive techniques outperform non-adaptive ones, which encouraged the authors to study the problem in a general case?
4. They didn't provide enough justification why the example in Section 5 (structured $\ell$_2 minimization with an oblivious channel)  is important in practice.
5. Since the main novelty of the work is to derive lower bounds for adaptive algorithms, I encourage the authors to discuss this more in section 3. It would be nice if authors provided a technical comparison with Archarya et al. (2021), and outlined how their method distinguishes itself.

I would like to thank the authors for their response. The authors addressed my concerns in the rebuttal. After reading other reviews and the authors' feedback, my assessment of the paper remains unchanged.

**Time Spent Reviewing:**

7 hours

---

> ### Author Response · Authors · 2021-08-08
> **Response to the review by Z9A8**
>
> We thank the reviewer for a thorough reading of our paper and many valuable comments. We will fix the typos pointed out by the reviewer. We respond to some of the other comments by the reviewer below.
>
> Response to 3: Generally speaking, adaptive protocols could lead to significant savings in data efficiency (and, in other settings and areas, are known to provide such savings). However, there would then be a tradeoff to consider, as implementing adaptive protocols may be more costly (in terms of space/time complexity) than nonadaptive ones. For this reason, most of the communication, privacy primitives developed for information-constrained optimization in the literature are nonadaptive in nature; but it’s unclear whether this was the right thing to do (in terms of that possible tradeoff curve). Our result shows that for some very popular function classes, this choice was the right one! There is no tradeoff to worry about, and it is enough to stick to nonadaptive protocols - one need not try to implement costly adaptive protocols.
>
>
> Response to 4: The specific example was chosen to point out that there may be structured optimization problems where adaptivity helps. While this was a theoretical, “proof-of-existence” example, one can think of practical optimization problems which have a similar structure. For instance, similar structural sparsity assumptions have been recently studied for the phase retrieval problem in signal processing, which, in turn, has applications to imaging systems such as diffraction imaging and X-ray crystallography; see [1] and the references therein.
>
> [1] Jagatap, Gauri, and Chinmay Hegde. "Fast, sample-efficient algorithms for structured phase retrieval." Proceedings of the 31st International Conference on Neural Information Processing Systems. 2017.
>
>
> Response to 5:  We build upon the results of Acharya et al. 2021 to derive lower bounds for information-constrained optimization. More specifically,  Acharya et al. 2021 focus on deriving lower bounds for estimation problems under information constraints.
> In our case, we derive lower bounds for information-constrained *optimization*. We do this by using Assouad’s method and show it is enough to bound the *average* mutual information (MI) between coordinates of the function parameter and the coded gradient output. Note that this average MI is smaller than the joint MI term, which appears in Fano’s method of Agarwal ‘11, and is, therefore, easier to upper bound. Bringing out this average MI in the lower bound using Assouad’s method instead of joint MI of Fano’s method is the key difference between our work and prior work. It is unclear if the joint MI  used for nonadaptive lower bounds can be bounded for adaptive procedures. To the best of our knowledge, no Assouad-type reduction for first-order stochastic optimization has been derived in this general form before. To bound the average MI term, we use the results from Acharya et al. 2021.
>
> Interestingly, even the application of MI bounds from Acharya et al. isn’t straightforward in some cases. For instance, for the oblivious sampling channel, the specific bounds derived on MI are new. Moreover, for the class of strongly-convex functions, the MI term needs to be processed nontrivially before one can apply the MI bounds from Acharya et al.

---

> > ### Comment · Reviewer_Z9A8 · 2021-08-25
> > **Acknowledging**
> >
> > I would like to thank the authors for their response. After reading other reviews and the authors' feedback, my assessment of the paper remains unchanged.

---

### Official Review · Reviewer_3YxU · 2021-07-16

**Rating:** 7
**Confidence:** 3

**Summary:**

In this paper, authors consider lower bounds for convex optimization where the stochastic gradient estimate is only partially observed. This sort of information constraint situation arises naturally when optimizing under local differential privacy, communication constraints, or randomized coordinate descent. The unique aspect of this paper is that they study *adaptive* information constraints, where the parts of the gradient that are revealed are chosen adaptively at each iteration by the algorithm, rather than fixed at the start of the algorithm.

The overall main result is that adaptively selecting gradient information does not provide any improvement on the worst case convergence rates, compared to non-adaptive selection. A proof sketch of this technique is provided at the end of the paper. An interesting positive result is a class of functions under which adaptively selecting gradient information does improve the convergence rate.

**Ethical Concerns:**

All ethical concerns are addressed.

**Limitations And Societal Impact:**

Yes.

**Main Review:**

The paper is extremely well written. My background is not in lower bounds, but I was able to understand the landscape of current work and the main contributions of the authors. Moreover, authors clearly described several technical aspects at a high level -- including obstructions for applying previously developed non-adaptive lower bound constructions to the adaptive setting. In this sense, the paper is extremely well written and a joy to read.

Because I do not work in the development of lower bounds, I cannot (with strong authority) comment on the relevance and novelty of the methods and results in the paper. However, based on the author’s description of the landscape of results, I believe that the results are timely and interesting. I was especially interested to see that the authors included a class of functions where adaptive selection of gradient information does provably improve over non-adaptive methods.

I recommend this paper for acceptance at NeurIPS 2021. I have found only one typo: In line 344, the output is referred to as $x_T$ but the display equation uses $\hat{x}$. I believe these should be the same.

**Time Spent Reviewing:**

2

---

> ### Author Response · Authors · 2021-08-08
> **Response to the review by 3YxU**
>
> We thank the reviewer for a thorough reading of our paper, their valuable comments, and for pointing out the typos. We will address the typos in the updated version of the paper.

---

### Official Review · Reviewer_bUwU · 2021-07-19

**Rating:** 6
**Confidence:** 4

**Summary:**


This paper concerns information theoretic lower bounds on
information-constrained first order optimization: where
the access to gradients from the first order oracle
is through a possibly adaptive noisy channel. As
examples of this model, they introduce privacy (via
differentially private channels), communication (via
r-bit channels) and computation (via coordinate-wise restricting
channels) constraints. For all of these cases, the paper
proves lower bounds showing (for most parameters considered)
that non-adaptive algorithms are nearly tight. They also
demonstrate a simple group-sparse mean estimation problem
where adaptively processing the gradients strictly
improves over the rate, in terms of dimension/sparsty dependence,
non-adaptive strategies.

**Limitations And Societal Impact:**

The paper being theoretical in nature does not have currently foreseeable societal impact

**Main Review:**


The paper is overall well-written, the contributions are clearly laid
out and the question is of traditional interest to the NeurIPS community.
The only potential weakness of the work is that its contributions are
largley lower bounds, and thereby not algorithmic.
Based on this, I would recommend its acceptance to the program.

Minor comments:
1. L.168-170 do not strictly follow since the set of vectors
E(hatg(x)) need not fully cover the subgradient. However, the
lipschitzness argument does seem fine.
2. Thms 1, 2 (and possibly later): why not include p \in (1, 2)?
The bound is not harder or more complicated to state, it is the same as p=1 from the
supplement.
3. Similar to point 2: why is this range for p not covered in the algorithmic
literature? Any particular reason this regime is not interesting for applications?
4. Minor grammar errors in L.258 266
5. The notation in the mutual information (\vee) is possibly non-standard. Please
either use I(X; Y) or explain this earlier in the text, particularly since you use
the same as \min, \max operators in the bounds.


**Time Spent Reviewing:**

3

---

> ### Author Response · Authors · 2021-08-08
> **Response to the review by bUwU**
>
> We thank the reviewer for a thorough reading of our paper and many valuable comments. We will fix the typos pointed out by the reviewer. We respond to some of the other comments by the reviewer below.
>
> "The only potential weakness of the work is that its contributions are largely lower bounds, and thereby not algorithmic."
> We agree with the reviewer that most of our results are lower bounds (except for a couple of new upper bounds; more on this later.). However, we emphasize that our lower bounds thus establish the optimality of various known algorithms from the literature, which, until now, in many cases were not known to be optimal in the interactive setting. In this sense, while our work does not provide new, faster algorithms for those tasks, it shows that they cannot exist: there is no hope to obtain significantly faster algorithms than the ones already existing.
>
>  As minor points, we note that our upper bound for the Convex and $\ell_1$ Lipschitz function families follow by developing an entirely new algorithm; see Algorithm 2 in the supplementary material. Moreover, the Adaptive coordinate descent method (ACD) is also novel for the structured optimization problem we describe in Section 3.4.
>
>
> Minor Comment 1: Good catch; thanks for pointing it out. We had already fixed this in the updated version of the paper (see the supplementary file).
>
> Minor Comment 3 -4:  The upper bounds for p in (1, 2) are by an off by a sublinear factor in d. We choose to provide the lower bounds for these parameters in the supplementary material and not the main paper because it is not clear if the bounds are tight in this case. Generally speaking, deriving tight upper bounds for all parameters (both the Lipschitz continuity parameter and the information constraints parameter ) is nontrivial, as tight bounds are unknown even in simpler nonadaptive settings. This difficulty could also be perhaps the reason this regime was not considered previously.
>
> Minor Comment 5: Thank you. We will add a clarifying text.

---

### Author Response · Authors · 2021-08-10
**Thank you to all the reviewers!**

We thank the reviewers for putting in the effort to review the paper and for many positive, encouraging comments about our results!

 We also thank the reviewers for raising a few concerns and for asking some interesting questions. We believe we have been able to address the concerns of the reviewers. And as one can conclude from our response, most of the concerns raised by the reviewers were minor and did not take anything away from the paper's main results.

We also believe that our results further push the boundaries of distributed optimization research, which will guide the design of various information-constrained primitives for optimization and inspire lower bounds for different function families and information constraints.

Therefore, we hope that the reviewers are satisfied with our response, see the merit of our main results, and kindly request that they consider increasing their scores.

---

### Decision · Program_Chairs · 2021-09-27

**Decision:**

Accept (Poster)

**Comment:**

The reviewers agree that this generally a good paper although not entirely without (minor) flaws. Please take the reviewers comments in consideration when preparing a revision. The answers provided by the authors were given due consideration.